# Certainty Equivalence is Efficient for Linear Quadratic Control

**Horia Mania**
University of California, Berkeley
hmania@berkeley.edu

**Stephen Tu**
University of California, Berkeley
stephentu@berkeley.edu

**Benjamin Recht**
University of California, Berkeley
brecht@berkeley.edu

## Abstract

We study the performance of the *certainty equivalent controller* on Linear Quadratic (LQ) control problems with unknown transition dynamics. We show that for both the fully and partially observed settings, the sub-optimality gap between the cost incurred by playing the certainty equivalent controller on the true system and the cost incurred by using the optimal LQ controller enjoys a fast statistical rate, scaling as the *square* of the parameter error. To the best of our knowledge, our result is the first sub-optimality guarantee in the partially observed Linear Quadratic Gaussian (LQG) setting. Furthermore, in the fully observed Linear Quadratic Regulator (LQR), our result improves upon recent work by Dean et al. [11], who present an algorithm achieving a sub-optimality gap linear in the parameter error. A key part of our analysis relies on perturbation bounds for discrete Riccati equations. We provide two new perturbation bounds, one that expands on an existing result from Konstantinov et al. [25], and another based on a new elementary proof strategy.

## 1 Introduction

One of the most straightforward methods for controlling a dynamical system with unknown transitions is based on the *certainty equivalence principle*: a model of the system is fit by observing its time evolution, and a control policy is then designed by treating the fitted model as the truth [6]. Despite the simplicity of this method, it is challenging to guarantee its efficiency because small modeling errors may propagate to large, undesirable behaviors on long time horizons. As a result, most work on controlling systems with unknown dynamics has explicitly incorporated robustness against model uncertainty [11, 12, 23, 30, 41, 42].

In this work, we show that for the standard baseline of controlling an unknown linear dynamical system with a quadratic objective function known as Linear Quadratic (LQ) control, certainty equivalent control synthesis achieves *better* cost than prior methods that account for model uncertainty. Our results hold for both the fully observed Linear Quadratic Regulator (LQR) and the partially observed Linear Quadratic Gaussian (LQG) setting. For offline control, where one collects some data and then designs a fixed control policy to be run on an infinite time horizon, we show that the gap between the performance of the certainty equivalent controller and the optimal control policy scales *quadratically* with the error in the model parameters for both LQR and LQG. To the best of our knowledge, we provide the first sub-optimality guarantee for LQG. Moreover, in the LQR setting our work improves upon the recent result of Dean et al. [11], who present an algorithm that achieves a sub-optimality gap linear in the parameter error. In the case of online LQR control, where one

adaptively improves the control policy as new data comes in, our offline result implies that a simple, polynomial time algorithm using $\varepsilon$-greedy exploration suffices for nearly optimal $\widetilde{\mathcal{O}}(\sqrt{T})$ regret.

## 2 Main Results for the Linear Quadratic Regulator

An instance of the linear quadratic regulator (LQR) is defined by four matrices: two matrices $A_\star \in \mathbb{R}^{n \times n}$ and $B_\star \in \mathbb{R}^{n \times d}$ that define the linear dynamics and two positive semidefinite matrices $Q \in \mathbb{R}^{n \times n}$ and $R \in \mathbb{R}^{d \times d}$ that define the cost function. Given these matrices, the goal of LQR is to solve the optimization problem

$$\min_{\mathbf{u}_0, \mathbf{u}_1, \dots} \lim_{T \to \infty} \mathbb{E} \left[ \frac{1}{T} \sum_{t=0}^{T} \mathbf{x}_t^\top Q \mathbf{x}_t + \mathbf{u}_t^\top R \mathbf{u}_t \right] \text{ s.t. } \mathbf{x}_{t+1} = A_\star \mathbf{x}_t + B_\star \mathbf{u}_t + \mathbf{w}_t, \tag{1}$$

where $\mathbf{x}_t$, $\mathbf{u}_t$ and $\mathbf{w}_t$ denote the state, input (or action), and noise at time $t$, respectively. The expectation is over the initial state $\mathbf{x}_0 \sim \mathcal{N}(0, I_n)$ and the i.i.d. noise $\mathbf{w}_t \sim \mathcal{N}(0, \sigma_w^2 I_n)$. When the problem parameters $(A_\star, B_\star, Q, R)$ are known the optimal policy is given by linear feedback, $\mathbf{u}_t = K_\star \mathbf{x}_t$, where $K_\star = -(R + B_\star^\top P_\star B_\star)^{-1} B_\star^\top P_\star A_\star$ where $P_\star$ is the (positive definite) solution to the discrete Riccati equation

$$P_\star = A_\star^\top P_\star A_\star - A_\star^\top P_\star B_\star (R + B_\star^\top P_\star B_\star)^{-1} B_\star^\top P_\star A_\star + Q \tag{2}$$

and can be computed efficiently [4, see e.g.]. Problem (1) considers an average cost over an infinite horizon. The optimal controller for the finite horizon variant is also static and linear, but time-varying. The LQR solution in this case can be computed efficiently via dynamic programming.

In this work we are interested in the control of a linear dynamical system with unknown transition parameters $(A_\star, B_\star)$ based on estimates $(\widehat{A}, \widehat{B})$. The cost matrices $Q$ and $R$ are assumed known. We analyze the *certainty equivalence approach*: use the estimates $(\widehat{A}, \widehat{B})$ to solve the optimization problem (1) while disregarding the modeling error, and use the resulting controller on the true system $(A_\star, B_\star)$. We interchangeably refer to the resulting policy as the *certainty equivalent controller* or, following Dean et al. [11], the *nominal controller*. We denote by $\widehat{P}$ the solution to the Riccati equation (2) associated with the parameters $(\widehat{A}, \widehat{B})$ and let $\widehat{K}$ be the corresponding controller. We denote by $J(A, B, K)$ the cost (1) obtained by using the actions $\mathbf{u}_t = K\mathbf{x}_t$ on the system $(A, B)$, and we use $\widehat{J}$ and $J_\star$ to denote $J(A_\star, B_\star, \widehat{K})$ and $J(A_\star, B_\star, K_\star)$, respectively.

Let $\varepsilon \geq 0$ such that $\|A_\star - \widehat{A}\| \leq \varepsilon$ and $\|B_\star - \widehat{B}\| \leq \varepsilon$. (Here and throughout this work we use $\|\cdot\|$ to denote the Euclidean norm for vectors as well as the spectral (operator) norm for matrices.) Dean et al. [11] introduced a robust controller that achieves $\widehat{J} - J_\star \leq C_1(A_\star, B_\star, Q, R)\varepsilon$ for some complexity term $C_1(A_\star, B_\star, Q, R)$ that depends on the problem parameters. We show that the nominal controller $\mathbf{u}_t = \widehat{K}\mathbf{x}_t$ achieves $\widehat{J} - J_\star \leq C_2(A_\star, B_\star, Q, R)\varepsilon^2$. Both results require $\varepsilon$ to be sufficiently small (as a function of the problem parameters) and it is important to note that $\varepsilon$ must be much smaller for the nominal controller to be guaranteed to stabilize the system than for the robust controller proposed by Dean et al. [11]. However, our result shows that once the estimation error $\varepsilon$ is small enough, the nominal controller performs better: the sub-optimality gap scales as $\mathcal{O}(\varepsilon^2)$ versus $\mathcal{O}(\varepsilon)$. Both the more stringent requirement on $\varepsilon$ and better performance of nominal control compared to robust control, when the estimation error is sufficiently small, were observed empirically by Dean et al. [11].

Before we can formally state our result we need to introduce a few more concepts and assumptions. It is common to assume that the cost matrices $Q$ and $R$ are positive definite. Under an additional observability assumption, this condition can be relaxed to $Q$ being positive semidefinite.

**Assumption 1.** *The cost matrices $Q$ and $R$ are positive definite. Since scaling both $Q$ and $R$ does not change the optimal controller $K_\star$, we can assume without loss of generality that $\underline{\sigma}(R) \geq 1$, where $\underline{\sigma}(\cdot)$ denotes the minimum singular value.*

A square matrix $M$ is *stable* if its spectral radius $\rho(M)$ is (strictly) smaller than one. Recall that the spectral radius is defined as $\rho(M) = \max\{|\lambda| : \lambda \text{ is an eigenvalue of } M\}$. A linear dynamical system $(A, B)$ in feedback with $K$ is fully described by the *closed loop matrix* $A + BK$. More precisely, in this case $\mathbf{x}_{t+1} = (A + BK)\mathbf{x}_t + \mathbf{w}_t$. For a static linear controller $\mathbf{u}_t = K\mathbf{x}_t$ to achieve finite LQR cost it is necessary and sufficient that the closed loop matrix is stable.

In order to quantify the growth or decay of powers of a square matrix $M$, we define

$$\tau(M, \rho) := \sup \left\{ \|M^k\| \rho^{-k} : k \geq 0 \right\}. \tag{3}$$

In other words, $\tau(M, \rho)$ is the smallest value such that $\|M^k\| \leq \tau(M, \rho) \rho^k$ for all $k \geq 0$. We note that $\tau(M, \rho)$ might be infinite, depending on the value of $\rho$, and it is always greater or equal than one. If $\rho$ is larger than $\rho(M)$, we are guaranteed to have a finite $\tau(M, \rho)$ (this is a consequence of Gelfand's formula). In particular, if $M$ is a stable matrix, we can choose $\rho < 1$ such that $\tau(M, \rho)$ is finite. Also, we note that $\tau(M, \rho)$ is a decreasing function of $\rho$; if $\rho \geq \|M\|$, we have $\tau(M, \rho) = 1$. At a high level, the quantity $\tau(M, \rho)$ measures the degree of transient response of the linear system $\mathbf{x}_{t+1} = M \mathbf{x}_t + \mathbf{w}_t$. In particular, when $M$ is stable, $\tau(M, \rho)$ can be upper bounded by the $\mathcal{H}_\infty$-norm of the system defined by $M$, which is the $\ell_2$ to $\ell_2$ operator norm of the system and a fundamental quantity in robust control [see 40, for more details].

Throughout this work we use the quantities $\Gamma_\star := 1 + \max\{\|A_\star\|, \|B_\star\|, \|P_\star\|, \|K_\star\|\}$ and $L_\star := A_\star + B_\star K_\star$. We use $\Gamma_\star$ as a uniform upper bound on the spectral norms of the relevant matrices for the sake of algebraic simplicity. We are ready to state our meta theorem. The proofs for all the results can be found in the full version of the paper [28].

**Theorem 1.** *Suppose $d \leq n$. Let $\gamma > 0$ such that $\rho(L_\star) \leq \gamma < 1$. Also, let $\varepsilon > 0$ such that $\|\widehat{A} - A_\star\| \leq \varepsilon$ and $\|\widehat{B} - B_\star\| \leq \varepsilon$ and assume $\|\widehat{P} - P_\star\| \leq f(\varepsilon)$ for some function $f$ such that $f(\varepsilon) \geq \varepsilon$. Then, under Assumption 1 the certainty equivalent controller $\mathbf{u}_t = \widehat{K} \mathbf{x}_t$ achieves*

$$\widehat{J} - J_\star \leq 200 \, \sigma_w^2 \, d \, \Gamma_\star^9 \, \frac{\tau(L_\star, \gamma)^2}{1 - \gamma^2} f(\varepsilon)^2, \tag{4}$$

*as long as $f(\varepsilon)$ is small enough so that the right hand side is smaller than $\sigma_w^2$.*

In Section 4 we present two upper bounds $f(\varepsilon)$ on $\|\widehat{P} - P_\star\|$: one based on a proof technique proposed by Konstantinov et al. [25] and one based on our direct approach. Both of these upper bounds satisfy $f(\varepsilon) = \mathcal{O}(\varepsilon)$ for $\varepsilon$ sufficiently small. For simplicity, in this section we only specialize our meta-theorem (Theorem 1) using the perturbation result from our direct approach.

To state a specialization of Theorem 1 we need a few more concepts. A linear system $(A, B)$ is called *controllable* when the *controllability matrix* $\begin{bmatrix} B & AB & A^2B & \dots & A^{n-1}B \end{bmatrix}$ has full row rank. Controllability is a fundamental concept in control theory; it states that there exists a sequence of inputs to the system $(A, B)$ that moves it from any starting state to any final state in at most $n$ steps. In this work we quantify how controllable a linear system is. We denote, for any integer $\ell \geq 1$, the matrix $\mathcal{C}_\ell := \begin{bmatrix} B & AB & \dots & A^{\ell-1}B \end{bmatrix}$ and call the system $(\ell, \nu)$-*controllable* if the $n$-th singular value of $\mathcal{C}_\ell$ is greater or equal than $\nu$, i.e. $\underline{\sigma}(\mathcal{C}_\ell) = \sqrt{\lambda_{\min}\left(\mathcal{C}_\ell \mathcal{C}_\ell^\top\right)} \geq \nu$. Intuitively, the larger $\nu$ is, the less control effort is needed to move the system between two different states.

**Assumption 2.** *We assume the unknown system $(A_\star, B_\star)$ is $(\ell, \nu)$-controllable, with $\nu > 0$.*

Assumption 2 was used in a different context by Cohen et al. [9]. For any controllable system and any $\ell \geq n$ there exists $\nu > 0$ such that the system is $(\ell, \nu)$-controllable. Therefore, $(\ell, \nu)$-controllability is really not much stronger of an assumption than controllability. As $\ell$ grows minimum singular value $\underline{\sigma}(\mathcal{C}_\ell)$ also grows and therefore a larger $\nu$ can be chosen so that the system is still $(\ell, \nu)$ controllable.

Note that controllability is not necessary for LQR to have a well-defined solution: the weaker requirement is that of *stabilizability*, in which there exists a feedback matrix $K$ so that $A_\star + B_\star K$ is stable. The result of Dean et al. [11] only requires stabilizability. While our upper bound on $\|\widehat{P} - P_\star\|$ requires controllability, the result of Konstantinov et al. [25] only requires stabilizability. However, our upper bound on $\|\widehat{P} - P_\star\|$ is sharper for some classes of systems (see Section 4). A direct plug in of our perturbation result, presented in Section 4, into Theorem 1 yields the following guarantee.

**Theorem 2.** *Suppose that $d \leq n$. Let $\rho$ and $\gamma$ be two real values such that $\rho(A_\star) \leq \rho$ and $\rho(L_\star) \leq \gamma < 1$. Also, let $\varepsilon > 0$ such that $\|\widehat{A} - A_\star\| \leq \varepsilon$ and $\|\widehat{B} - B_\star\| \leq \varepsilon$ and define $\beta = \max\{1, \varepsilon\tau(A_\star, \rho) + \rho\}$. Under Assumptions 1 and 2, the certainty equivalent controller $\mathbf{u}_t = \widehat{K} \mathbf{x}_t$ satisfies the suboptimality gap*

$$\widehat{J} - J_\star \leq \mathcal{O}(1) \, \sigma_w^2 \, d \, \ell^5 \, \Gamma_\star^{15} \, \tau(A_\star, \rho)^6 \beta^{4(\ell-1)} \frac{\tau(L_\star, \gamma)^2}{1 - \gamma^2} \frac{\max\{\|Q\|^2, \|R\|^2\}}{\min\{\underline{\sigma}(Q)^2, \underline{\sigma}(R)^2\}} \left(1 + \frac{1}{\nu}\right)^2 \varepsilon^2, \tag{5}$$

*as long as the right hand side is smaller than $\sigma_w^2$. Here, $\mathcal{O}(1)$ denotes a universal constant.*

The exact form of Equation 5, such as the polynomial dependence on $\ell, \Gamma_\star$, etc, can be improved at the expense of conciseness of the expression. In our proof we optimized for the latter. The factor $\max\{\|Q\|^2, \|R\|^2\}/\min\{\underline{\sigma}(Q)^2, \underline{\sigma}(R)^2\}$ is the squared condition number of the cost function, a natural quantity in the context of the optimization problem (1), which can be seen as an infinite dimensional quadratic program with a linear constraint. The term $\frac{\tau(L_\star,\gamma)^2}{1-\gamma^2}$ quantifies the rate at which the optimal controller drives the state towards zero. Generally speaking, the less stable the optimal closed loop system is, the larger this term becomes.

An interesting trade-off arises between the factor $\ell^5 \beta^{4(\ell-1)}$ (which arises from upper bounding perturbations of powers of $A_\star$ on a time interval of length $\ell$) and the factor $\nu$ (the lower bound on $\underline{\sigma}(\mathcal{C}_\ell)$), which is increasing in $\ell$. Hence, the parameter $\ell$ should be seen as a free-parameter that can be tuned to minimize the right hand side of (5). Now, we specialize Theorem 2 to a few cases.

**Case: $A_\star$ is contractive, i.e. $\|A_\star\| < 1$.** In this case, we can choose $\rho = \|A_\star\|$ and $\varepsilon$ small enough so that $\varepsilon \leq 1 - \|A_\star\|$. Then, (5) simplifies to:

$$\widehat{J} - J_\star \leq \mathcal{O}(1)\, d\, \sigma_w^2\, \ell^5\, \Gamma_\star^{15} \frac{\tau(L_\star,\gamma)^2}{1-\gamma^2} \frac{\max\{\|Q\|^2, \|R\|^2\}}{\min\{\underline{\sigma}(Q)^2, \underline{\sigma}(R)^2\}} \left(1 + \frac{1}{\nu}\right)^2 \varepsilon^2 \, .$$

**Case: $B_\star$ has rank $n$.** In this case, we can choose $\ell = 1$. Then, (5) simplifies to:

$$\widehat{J} - J_\star \leq \mathcal{O}(1)\, d\, \sigma_w^2\, \Gamma_\star^{15} \tau(A_\star,\rho)^6 \frac{\tau(L_\star,\gamma)^2}{1-\gamma^2} \frac{\max\{\|Q\|^2, \|R\|^2\}}{\min\{\underline{\sigma}(Q)^2, \underline{\sigma}(R)^2\}} \left(1 + \frac{1}{\nu}\right)^2 \varepsilon^2 \, .$$

## 2.1 Comparison to Theorem 4.1 of Dean et al. [11].

Dean et al. [11] show that when their robust synthesis procedure is run with estimates $(\widehat{A}, \widehat{B})$ satisfying $\max\{\|\widehat{A} - A_\star\|, \|\widehat{B} - B_\star\|\} \leq \varepsilon \leq [5(1 + \|K_\star\|)\Psi_\star]^{-1}$, the resulting controller satisfies:

$$\widehat{J} - J_\star \leq 10(1 + \|K_\star\|)\Psi_\star J_\star \varepsilon + \mathcal{O}(\varepsilon^2) \, . \tag{6}$$

Here, the quantity $\Psi_\star := \sup_{z \in \mathbb{T}} \|(zI_n - L_\star)^{-1}\|$ is the $\mathcal{H}_\infty$-norm of the optimal closed loop system $L_\star$. In order to compare Equation 6 to Equation 5, we upper bound the quantity $\Psi_\star$ in terms of $\tau(L_\star,\gamma)$ and $\gamma$. In particular, by a infinite series expansion of the inverse $(zI_n - L_\star)^{-1}$ we can show $\Psi_\star \leq \frac{\tau(L_\star,\gamma)}{1-\gamma}$. Also, we have $J_\star = \sigma_w^2 \operatorname{tr}(P_\star) \leq \sigma_w^2 n \Gamma_\star$. Therefore, Equation 6 gives us that:

$$\widehat{J} - J_\star \leq \mathcal{O}(1) n \sigma_w^2 \Gamma_\star^2 \frac{\tau(L_\star,\gamma)}{1-\gamma} \varepsilon + \mathcal{O}(\varepsilon^2) \, .$$

We see that the dependence on the parameters $\Gamma_\star$ and $\tau(L_\star,\gamma)$ is significantly milder compared to Equation 5. Furthermore, this upper bound is valid for larger $\varepsilon$ than the upper bound given in Theorem 2. Comparing these upper bound suggests that there is a price to pay for obtaining a fast rate, and that in regimes of moderate uncertainty (moderate size of $\varepsilon$), being robust to model uncertainty is important. This observation is supported by the empirical results of Dean et al. [11].

A similar trade-off between slow and fast rates arises in the setting of first-order convex stochastic optimization. The convergence rate $\mathcal{O}(1/\sqrt{T})$ of the stochastic gradient descent method can be improved to $\mathcal{O}(1/T)$ under a strong convexity assumption. However, the performance of stochastic gradient descent, which can achieve a $\mathcal{O}(1/T)$ rate, is sensitive to poorly estimated problem parameters [29]. Similarly, in the case of LQR, the nominal controller achieves a fast rate, but it is much more sensitive to estimation error than the robust controller of Dean et al. [11].

**End-to-end guarantees.** Theorem 2 can be combined with finite sample learning guarantees (e.g. [11, 15, 33, 34]) to obtain an end-to-end guarantee similar to Proposition 1.2 of Dean et al. [11]. In general, estimating the transition parameters from $N$ samples yields an estimation error that scales as $\mathcal{O}(1/\sqrt{N})$. Therefore, Theorem 2 implies that $\widehat{J} - J_\star \leq \mathcal{O}(1/N)$ instead of the $\widehat{J} - J_\star \leq \mathcal{O}(1/\sqrt{N})$ rate from Proposition 1.2 of Dean et al. [11]. This is similar to the case of linear regression, where $\mathcal{O}(1/\sqrt{N})$ estimation error for the parameters translates to a $\mathcal{O}(1/N)$ *fast rate* for prediction error. Furthermore, Simchowitz et al. [34] and Sarkar and Rakhlin [33] showed

that faster estimation rates are possible for some linear dynamical systems. Theorem 2 translates such rates into control suboptimality guarantees in a transparent way.

Our result explains the behavior observed in Figure 4 of Dean et al. [11]. The authors propose two procedures for synthesizing robust controllers for LQR with unknown transitions: one which guarantees robustness of the performance gap $\widehat{J} - J_\star$, and one which only guarantees the stability of the closed loop system. Dean et al. [11] observed that the latter performs better in the small estimation error regime, which happens because the robustness constraint of the synthesis procedure becomes inactive when the estimation error is small enough. Then, the second robust synthesis procedure effectively outputs the certainty equivalent controller, which we now know to achieve a fast rate.

## 2.2 Nearly optimal $\widetilde{\mathcal{O}}(\sqrt{T})$ regret in the adaptive setting

The regret formulation of adaptive LQR was first proposed by Abbasi-Yadkori and Szepesvári [1]. The task is to design an adaptive algorithm $\{\mathbf{u}_t\}_{t \geq 0}$ to minimize regret, as defined by $\mathsf{Regret}(T) := \sum_{t=1}^{T} \mathbf{x}_t^\top Q \mathbf{x}_t + \mathbf{u}_t^\top R \mathbf{u}_t - T J_\star$. Abbasi-Yadkori and Szepesvári [1] study the performance of optimism in the face of uncertainty (OFU) and show that it has $\widetilde{\mathcal{O}}(\sqrt{T})$ regret, which is nearly optimal for this problem formulation. However, the OFU algorithm requires repeated solutions to a non-convex optimization problem for which no known efficient algorithm exists.

To deal with the computational issues of OFU, Dean et al. [12] propose to analyze the behavior of $\varepsilon$-greedy exploration using the suboptimality gap results of Dean et al. [11]. In the context of continuous control, $\varepsilon$-greedy exploration refers to the application of the control law $\mathbf{u}_t = \pi(\mathbf{x}_t, \mathbf{x}_{t-1}, ..., \mathbf{x}_0) + \eta_t$ with $\eta_t \sim \mathcal{N}(0, \sigma_{\eta,t}^2 I_d)$, where $\pi$ is the policy, updated in epochs, and $\sigma_{\eta,t}^2$ is the variance of the exploration noise. Dean et al. [12] set the variance of the exploration noise as $\sigma_{\eta,t}^2 \sim t^{-1/3}$, and show that their method achieves $\widetilde{\mathcal{O}}(T^{2/3})$ regret. They use epochs of size $2^i$ and decompose the regret roughly as $\mathsf{Regret}(T) = \mathcal{O}\left(T(\widehat{J} - J_\star) + T\sigma_{\eta,T}^2\right)$. Since the estimation error of the model parameters scales as $\mathcal{O}((\sigma_{\eta,T}\sqrt{T})^{-1})$, and since the suboptimality gap $\widehat{J} - J_\star$ of the robust controller is linear in the estimation error, we have $\mathsf{Regret}(T) = \mathcal{O}\left(\frac{\sqrt{T}}{\sigma_{\eta,T}} + T\sigma_{\eta,T}^2\right)$. Then, setting $\sigma_{\eta,t}^2 \sim t^{-1/3}$ balances these two terms and yields $\widetilde{\mathcal{O}}(T^{2/3})$ regret. However, Theorem 2, which states that the gap $\widehat{J} - J_\star$ for the nominal controller depends quadratically on the estimation rate, implies that online certainty equivalent control achieves $\mathsf{Regret}(T) = \mathcal{O}\left(\frac{1}{\sigma_{\eta,T}^2} + T\sigma_{\eta,T}^2\right)$. Here, the optimal variance of the exploration noise scales as $\sigma_{\eta,t}^2 \sim t^{-1/2}$, yielding $\widetilde{\mathcal{O}}(\sqrt{T})$ regret. We note that the observation that certainty equivalence coupled with $\varepsilon$-greedy exploration achieves $\widetilde{\mathcal{O}}(\sqrt{T})$ regret was first made by Faradonbeh et al. [16].

**Corollary 1.** *(Informal) $\varepsilon$-greedy exploration with exploration schedule $\sigma_{\eta,t}^2 \sim t^{-1/2}$ combined with certainty equivalent control yields an adaptive LQR algorithm with regret bounded as $\widetilde{\mathcal{O}}(\sqrt{T})$.*

## 3 Main Results for the Linear Quadratic Gaussian Problem

Now we consider partially observable systems. In this case the system dynamics have the form:

$$\mathbf{x}_{t+1} = A_\star \mathbf{x}_t + B_\star \mathbf{u}_t + \mathbf{w}_t , \quad \mathbf{w}_t \sim \mathcal{N}(0, \sigma_w^2 I) , \tag{7a}$$

$$\mathbf{y}_t = C_\star \mathbf{x}_t + \mathbf{v}_t , \quad \mathbf{v}_t \sim \mathcal{N}(0, \sigma_v^2 I) . \tag{7b}$$

In (7), only the output process $\mathbf{y}_t$ is observed. The LQG problem is defined as[1]:

$$\min_{\mathbf{u}_0, \mathbf{u}_1, ...} \lim_{T \to \infty} \mathbb{E}\left[\frac{1}{T} \sum_{t=0}^{T} \mathbf{y}_t^\top Q \mathbf{y}_t + \mathbf{u}_t^\top R \mathbf{u}_t\right] \quad \text{s.t. (7a), (7b) .} \tag{8}$$

Here, the input $\mathbf{u}_t$ is allowed to depend on the history[2] $\mathcal{H}_t := (\mathbf{u}_0, ..., \mathbf{u}_{t-1}, \mathbf{y}_0, ..., \mathbf{y}_{t-1})$. The optimal solution to (8) is to set $\mathbf{u}_t = K_\star \widehat{\mathbf{x}}_t$, with $K_\star$ the optimal LQR solution to $(A_\star, B_\star, C_\star^\mathsf{T} Q C_\star, R)$ and $\widehat{\mathbf{x}}_t := \mathbb{E}[\mathbf{x}_t | \mathcal{H}_t]$. The MSE estimate $\widehat{\mathbf{x}}_t$ can be solved efficiently via Kalman filtering:

$$\widehat{\mathbf{x}}_{t+1} = A_\star \widehat{\mathbf{x}}_t + B_\star \mathbf{u}_t + L_\star (\mathbf{y}_t - C_\star \widehat{\mathbf{x}}_t) \,, \tag{9a}$$

$$L_\star = -A_\star \Sigma_\star C_\star^\mathsf{T} (C_\star \Sigma_\star C_\star^\mathsf{T} + V)^{-1} \,, \tag{9b}$$

$$\Sigma_\star = A_\star \Sigma_\star A_\star^\mathsf{T} + \sigma_w^2 I - A_\star \Sigma_\star C_\star^\mathsf{T} (C_\star \Sigma_\star C_\star^\mathsf{T} + \sigma_v^2 I)^{-1} C_\star \Sigma_\star A_\star^\mathsf{T} \,. \tag{9c}$$

There is an inherent ambiguity in the dynamics (7a)-(7b) which makes LQG more delicate than LQR. In particular, for any invertible $T$, the LQG problem (8) with parameters $(A_\star, B_\star, C_\star, Q, R)$ is equivalent to the LQG problem with parameters $(TA_\star T^{-1}, TB_\star, C_\star T^{-1}, Q, R)$ and appropriately rescaled noise processes. To deal with this ambiguity, we assume that we have estimates $(\widehat{A}, \widehat{B}, \widehat{C}, \widehat{L})$ such that there exists an unitary $T$ such that:

$$\max\{\|\widehat{A} - TA_\star T^{-1}\|, \|\widehat{B} - TB_\star\|, \|\widehat{C} - C_\star T^{-1}\|, \|\widehat{L} - TL_\star\|\} \le \varepsilon \,. \tag{10}$$

Recent work [32, 35, 38] has shown how to obtain this style of estimates with guarantees from input/output data. As in Section 2, we assume that the cost matrices $(Q, R)$ are known. Then, we study the performance of the certainty equivalence controller defined by:

$$\widehat{\mathbf{x}}_{t+1} = \widehat{A} \widehat{\mathbf{x}}_t + \widehat{B} \mathbf{u}_t + \widehat{L} (\mathbf{y}_t - \widehat{C} \widehat{\mathbf{x}}_t) \,, \quad \mathbf{u}_t = \widehat{K} \widehat{\mathbf{x}}_t \,, \quad \widehat{K} = \mathsf{LQR}(\widehat{A}, \widehat{B}, \widehat{C}^\mathsf{T} Q \widehat{C}, R) \,. \tag{11}$$

Similarly to Theorem 1 for LQR, we state a meta theorem for LQG. Unlike Theorem 1, however, we need a stronger type of Riccati perturbation guarantee which also allows for perturbation of the $Q$ matrix. Specifically, we suppose there exists $\gamma_0$ such that for any $\gamma \le \gamma_0$ and $(\widehat{A}, \widehat{B}, \widehat{Q})$ with $\max\{\|\widehat{A} - A\|, \|\widehat{B} - B\|, \|\widehat{Q} - Q\|\} \le \gamma$ the solutions $P$ and $\widehat{P}$ of the Riccati equations with parameters $(A, B, Q, R)$ and $(\widehat{A}, \widehat{B}, \widehat{Q}, R)$ satisfy

$$\|P - \widehat{P}\| \le f(\gamma) \,, \tag{12}$$

for an increasing function $f$ with $f(\gamma) \ge \gamma$. The constant $\gamma_0$ and function $f$ are allowed to depend on the parameters $(A, B, Q, R)$. In Section 4, we present a perturbation bound (Proposition 1) that satisfies these properties. Similarly to Section 2, we define $\Gamma_\star := 1 + \max\{\|A_\star\|, \|B_\star\|, \|C_\star\|, \|K_\star\|, \|L_\star\|, \|P_\star\|\}$. The following theorem is our main result for LQG.

**Theorem 3.** *Suppose that $(A_\star, B_\star)$ is stabilizable, $(C_\star, A_\star)$ is observable, and that Assumption 1 holds. Let $\varepsilon$ be an upper bound on $\|\widehat{A} - TA_\star T^{-1}\|$, $\|\widehat{B} - TB_\star\|$, $\|\widehat{C} - C_\star T^{-1}\|$, and $\|\widehat{L} - TL_\star\|$ for some unitary transformation $T$. Suppose that assumption (12) holds with parameters $(TA_\star T^{-1}, TB_\star, T^{-\mathsf{T}} C_\star^\mathsf{T} Q C_\star T^{-1}, R)$ and that $\varepsilon$ is sufficiently small so that $3\|C_\star\|_+ \|Q\|_+ \varepsilon \le \gamma_0$ and $\bar{\varepsilon} \le 1$, where $\bar{\varepsilon} := \frac{7\Gamma_\star^3}{\underline{\sigma}(R)} f(3\|C_\star\|_+^2 \|Q\|_+ \varepsilon)$. Let $\widehat{K}$ be defined as in (11), and define $N_\star$ as*

$$N_\star := \begin{bmatrix} A_\star + B_\star K_\star & B_\star K_\star \\ 0 & A_\star - L_\star C_\star \end{bmatrix} \,, \tag{13}$$

*where the pair $(K_\star, L_\star)$ is optimal for the LQG problem defined by $(A_\star, B_\star, C_\star, Q, R)$. Let $\gamma > 0$ be such that $\rho(N_\star) < \gamma < 1$. Then as long as $\bar{\varepsilon} \le \frac{1-\gamma}{20\Gamma_\star \tau(N_\star, \gamma)}$, the interconnection of (11) with (7) using $(\widehat{A}, \widehat{B}, \widehat{C}, \widehat{K}, \widehat{L})$ is stable. Furthermore, the cost $J(\widehat{A}, \widehat{B}, \widehat{C}, \widehat{K}, \widehat{L})$ satisfies:*

$$J(\widehat{A}, \widehat{B}, \widehat{C}, \widehat{K}, \widehat{L}) - J_\star \le \mathcal{O}(1) \max\{\sigma_w^2, \sigma_v^2\} (\operatorname{tr}(C_\star^\mathsf{T} Q C_\star) + \operatorname{tr}(R)) \frac{\tau^6(N_\star, \gamma)}{(1-\gamma^2)^3} \Gamma_\star^6 \bar{\varepsilon}^2 \,.$$

The proof of Theorem 3 appears in Appendix F. We note that such a $\gamma$ exists since $\rho(N_\star) < 1$; by the stability and observability assumptions in Theorem 3, we have that both $A_\star + B_\star K_\star$ and $A_\star - L_\star C_\star$ are stable (c.f. Appendix E of Kailath et al. [24]). Theorem 3 is a meta-theorem showing how perturbation bounds on the solution of Riccati equations translates into suboptimality bounds on the performance of certainty equivalent control. Combining Theorem 3 with Proposition 1, we have the following explicit result, an analogue of Theorem 2 for LQG. To simplify notation we denote by $\mathrm{dare}(A, B, Q, R)$ the solution to the discrete algebraic Riccati equation defined by the parameters $A$, $B$, $Q$, and $R$.

**Theorem 4.** *Suppose that $(A_\star, B_\star)$ is stabilizable, $(C_\star, A_\star)$ is observable, and that Assumption 1 holds. Let $\varepsilon$ be an upper bound on $\|\widehat{A} - TA_\star T^{-1}\|$, $\|\widehat{B} - TB_\star\|$, $\|\widehat{C} - C_\star T^{-1}\|$, and $\|\widehat{L} - TL_\star\|$ for some unitary transformation $T$. Let $P_\star = \mathsf{dare}(A_\star, B_\star, C_\star^\mathsf{T} QC_\star, R)$ and suppose that $\underline{\sigma}(P_\star) \geq 1$. Let $N_\star$ be as in (13) and fix $\gamma$ such that $\rho(N_\star) < \gamma < 1$. As long as $\varepsilon$ satisfies $\varepsilon \leq \frac{(1-\gamma^2)^2}{\tau^4(N_\star, \gamma)} \frac{1}{\Gamma_\star^{11} \|Q\|}$, we have the following sub-optimality bound:*

$$J(\widehat{A}, \widehat{B}, \widehat{C}, \widehat{K}, \widehat{L}) - J_\star \leq \mathcal{O}(1) \max\{\sigma_w^2, \sigma_v^2\}(\operatorname{tr}(C_\star^\mathsf{T} QC_\star) + \operatorname{tr}(R)) \frac{\|Q\|^2}{\underline{\sigma}(R)^2} \Gamma_\star^{26} \frac{\tau^{10}(N_\star, \gamma)}{(1-\gamma^2)^5} \varepsilon^2.$$

Several remarks are in order. First, the assumption that $\underline{\sigma}(P_\star) \geq 1$ is without loss of generality, since we can always rescale $Q$ and $R$ without affecting the control solution. Next, we compare our results here to a classic result from Doyle [13], which states that there are no gain margins for LQG. We remark that the notion of a gain margin is a robustness property that holds uniformly over a class of perturbations of varying degree. Our results do not hold uniformly; we use quantities such as $\tau(N_\star, \gamma)$ and $\Gamma_\star$ to quantify how much mismatch a given LQG instance can tolerate.

## 4 Riccati Perturbation Theory

As discussed in Sections 2 and 3, a key piece of our analysis is bounding the solutions to discrete Riccati equations as we perturb the problem parameters. Specifically, we are interested in quantities $b, L$ such that $\|\widehat{P} - P_\star\| \leq L\varepsilon$ if $\varepsilon < b$, where $\varepsilon$ represents a bound on the perturbation. We note that it is not possible to find universal values $b, L$. Consider the systems $(A_\star, B_\star) = (1, \varepsilon)$ and $(\widehat{A}, \widehat{B}) = (1, 0)$; the latter system is not stabilizable and hence $\widehat{P}$ does not even exist. Therefore, $b$ and $L$ must depend on the system parameters.

While there is a long line of work analyzing perturbations of Riccati equations, we are not aware of any result that offers explicit and easily interpretable $b$ and $L$ for a fixed $(A_\star, B_\star, Q, R)$; see Konstantinov et al. [26] for an overview of this literature. In this section, we present two new results for Riccati perturbation which offer interpretable bounds. The first one expands upon the operator-theoretic proof of Konstantinov et al. [25]; its proof can be found in Appendix B.1. In this result we assume the cost matrix $Q$ can also be perturbed, which is needed for our LQG guarantee. In order to be consistent we denote the true cost by function by $Q_\star$ and the estimated one by $\widehat{Q}$.

**Proposition 1.** *Let $\gamma \geq \rho(L_\star)$ and also let $\varepsilon$ such that $\|\widehat{A} - A_\star\|$, $\|\widehat{B} - B_\star\|$, and $\|\widehat{Q} - Q_\star\|$ are at most $\varepsilon$. Let $\|\cdot\|_+ = \|\cdot\| + 1$. We assume that $R \succ 0$, $(A_\star, B_\star)$ is stabilizable, $(Q^{1/2}, A_\star)$ observable, and $\underline{\sigma}(P_\star) \geq 1$.*

$$\|\widehat{P} - P_\star\| \leq \mathcal{O}(1) \, \varepsilon \, \frac{\tau(L_\star, \gamma)^2}{1 - \gamma^2} \|A_\star\|_+^2 \|P_\star\|_+^2 \|B_\star\|_+ \|R^{-1}\|_+,$$

*as long as*

$$\varepsilon \leq \mathcal{O}(1) \frac{(1-\gamma^2)^2}{\tau(L_\star, \gamma)^4} \|A_\star\|_+^{-2} \|P_\star\|_+^{-2} \|B_\star\|_+^{-3} \|R^{-1}\|_+^{-2} \min\left\{\|L_\star\|_+^{-2}, \|P_\star\|_+^{-1}\right\}.$$

We note that the assumption $\underline{\sigma}(P_\star) \geq 1$ can be made without loss of generality when the other assumptions are satisfied. Since $R \succ 0$ and $(Q^{1/2}, A)$ observable, the value function matrix $P_\star$ is guaranteed to be positive definite. Then, by rescaling $Q$ and $R$ we can ensure that $\underline{\sigma}(P_\star) \geq 1$.

We now present our direct approach, which uses Assumption 2 to give a bound which is sharper for some systems $(A_\star, B_\star)$ then the one provided by Proposition 1. Recall that any controllable system is always $(\ell, \nu)$-controllable for some $\ell$ and $\nu$.

**Proposition 2.** *Let $\rho \geq \rho(A_\star)$ and also let $\varepsilon \geq 0$ such that $\|\widehat{A} - A_\star\| \leq \varepsilon$ and $\|\widehat{B} - B_\star\| \leq \varepsilon$. Let $\beta := \max\{1, \varepsilon\tau(A_\star, \rho) + \rho\}$. Under Assumptions 1 and 2 we have*

$$\|\widehat{P} - P_\star\| \leq 32 \, \varepsilon \, \ell^{\frac{5}{2}} \tau(A_\star, \rho)^3 \beta^{2(\ell-1)} \left(1 + \frac{1}{\nu}\right)(1 + \|B_\star\|)^2 \|P_\star\| \frac{\max\{\|Q\|, \|R\|\}}{\min\{\underline{\sigma}(R), \underline{\sigma}(Q)\}},$$

*as long as $\varepsilon$ is small enough so that the right hand side is smaller or equal than one.*

The proof of this result is deferred to Appendix B.2. We note that Proposition 2 can also be extended to handle perturbations in the cost matrix $Q$. Proposition 2 requires an $(\ell, \nu)$-controllable system $(A_\star, B_\star)$, whereas Proposition 1 only requires a stabilizable system, which is a milder assumption. However, Proposition 2 can offer a sharper guarantee. For example, consider the linear system with two dimensional states ($n = 2$) given by $A_\star = 1.01 \cdot I_2$ and $B_\star = \begin{bmatrix} 1 & 0 \\ 0 & \beta \end{bmatrix}$. Both $Q$ and $R$ are chosen to be the identity matrix $I_2$. This system $(A_\star, B_\star)$ is readily checked to be $(1, \beta)$-controllable. It is also straightforward to verify that as $\beta$ tends to zero, Proposition 1 gives a bound of $\|\widehat{P} - P_\star\| = \mathcal{O}(\varepsilon/\beta^4)$, whereas Proposition 2 gives a sharper bound of $\|\widehat{P} - P_\star\| = \mathcal{O}(\varepsilon/\beta^3)$.

## 5  Related Work

For the offline LQR batch setting, Fiechter [18] proved that the sub-optimality gap $\widehat{J} - J_\star$ scales as $\mathcal{O}(\varepsilon)$ for certainty equivalent control. A crucial assumption of his analysis is that the nominal controller stabilizes the true unknown system. We give bounds on when this assumption is valid. Recently, Dean et al. [11] proposed a robust controller synthesis procedure which takes model uncertainty into account and whose suboptimality gap scales as $\mathcal{O}(\varepsilon)$. Tu and Recht [39] show that the gap $\widehat{J} - J_\star$ of certainty equivalent control scales asymptotically as $\mathcal{O}(\varepsilon^2)$; we provide a non-asymptotic analogue of this result. Fazel et al. [17] and Malik et al. [27] analyze a model-free approach to policy optimization for LQR, in which the controller is directly optimized from sampled rollouts. Malik et al. [27] showed that, after collecting $N$ rollouts, a derivative free method achieves a discounted cost gap that scales as $\mathcal{O}(1/\sqrt{N})$ or $\mathcal{O}(1/N)$, depending on the oracle model used.

In the online LQR adaptive setting it is well understood that using the certainty equivalence principle without adequate exploration can result in a lack of parameter convergence [see e.g. 5]. Abbasi-Yadkori and Szepesvári [1] showed that optimism in the face of uncertainty (OFU), when applied to online LQR, yields $\widetilde{\mathcal{O}}(\sqrt{T})$ regret. Faradonbeh et al. [14] removed some un-necessary assumptions of the previous analysis. Ibrahimi et al. [22] showed that when the underlying system is sparse, the dimension dependent constants in the regret bound can be improved. The main issue with OFU for LQR is that there are no known computationally tractable ways of implementing it. In order to deal with this, both Dean et al. [12] and Abbasi-Yadkori et al. [2] propose polynomial time algorithms for adaptive LQR based on $\varepsilon$-greedy exploration which achieve $\widetilde{\mathcal{O}}(T^{2/3})$ regret. Only recently progress has been made on offering $\widetilde{\mathcal{O}}(\sqrt{T})$ regret guarantees for computationally tractable algorithms. Abeille and Lazaric [3] show that Thompson sampling achieves $\widetilde{\mathcal{O}}(\sqrt{T})$ (frequentist) regret for the case when the state and inputs are both scalars. In a Bayesian setting Ouyang et al. [31] showed that Thompson sampling achieves $\widetilde{\mathcal{O}}(\sqrt{T})$ *expected* regret. Faradonbeh et al. [16] argue that certainty equivalence control with an epsilon-greedy-like scheme achieves $\widetilde{\mathcal{O}}(\sqrt{T})$ regret, though their work does not provide any explicit dependencies on instance parameters. Finally, Cohen et al. [10] also give an efficient algorithm based on semidefinite programming that achieves $\widetilde{\mathcal{O}}(\sqrt{T})$ regret. Their main result requires the initial parameter error to scale as $\mathcal{O}(1/T^{1/4})$. While they propose a $\mathcal{O}(\sqrt{T})$ length warmup period to get around this, our analysis of $\varepsilon$-greedy control does not require $o_T(1)$ accuracy of the initial parameters. Moreover, there are specialized algorithms for solving Riccati equations that are more efficient than general semidefinite programming solvers.

The literature for LQG is less complete, with most of the focus on the estimation side. Hardt et al. [19] show that gradient descent can be used to learn a model with good predictive performance, under strong technical assumptions on the $A$ matrix. A line of work [20, 21] has focused on using spectral filtering techniques to learn a predictive model with low regret. Beyond predictive performance, several works [32, 35, 38] show how to learn the system dynamics up to a similarity transform from input/output data. Finally, we remark that Boczar et al. [8] give sub-optimality guarantees for output-feedback of a single-input-single-output (SISO) linear system with no process noise.

A key part of our analysis involves bounding the perturbation of solutions to the discrete algebraic Riccati equation. While there is a rich line of work studying perturbations of Riccati equations [25, 26, 36, 37], the results in the literature are either asymptotic in nature or difficult to use and interpret. We clarify the operator-theoretic result of Konstantinov et al. [25] and provide an explicit upper bound on the perturbation based on their proof strategy. Also, we take a new direct approach

and use an extended notion of controllability to give a constructive and simpler result. While the result of Konstantinov et al. [25] applies more generally to systems that are stabilizable, we give examples of linear systems for which our new perturbation result is tighter.

Finally, while we focus on a continuous control problem, we note that the performance of certainty equivalence had been studied in the context of tabular MDPs, e.g. Azar et al. [7] derived matching upper and lower bounds on the performance of value-iteration and policy-iteration that use estimated transition probabilities.

## 6 Conclusion

Though a naïve Taylor expansion suggests that the fast rates we derive here must be achievable, precisely computing such rates has been open since the 80s. All of the pieces we used here have existed in the literature for some time, and perhaps it has just required a bit of time to align contemporary rate-analyses in learning theory with earlier operator theoretic work in optimal control. There remain many possible extensions to this work. The robust control approach of Dean et al. [11] applies to many different objective functions besides quadratic costs, such as $\mathcal{H}_\infty$ and $\mathcal{L}_1$ control. It would be interesting to know whether fast rates for control are possible for other objective functions. Finally, determining the optimal minimax rate for both LQR and LQG would allow us to understand the tradeoffs between nominal and robust control at a more fine grained level.

## Acknowledgements

We thank the anonymous reviewers for their valuable feedback. We also thank Elad Hazan and Martin Wainwright, who both independently asked whether or not it was possible to show a fast rate for LQR. As part of the RISE lab, HM is generally supported in part by NSF CISE Expeditions Award CCF-1730628, DHS Award HSHQDC-16-3-00083, and gifts from Alibaba, Amazon Web Services, Ant Financial, CapitalOne, Ericsson, GE, Google, Huawei, Intel, IBM, Microsoft, Scotiabank, Splunk and VMware. ST is supported by a Google PhD fellowship. BR is generously supported in part by ONR awards N00014-17-1-2191, N00014-17-1-2401, and N00014-18-1-2833, the DARPA Assured Autonomy (FA8750-18-C-0101) and Lagrange (W911NF-16-1-0552) programs, a Siemens Futuremakers Fellowship, and an Amazon AWS AI Research Award.

## Footnotes

[1]Note that many texts define the LQG cost in terms of $\mathbf{x}_t^\top Q \mathbf{x}_t$ instead of $\mathbf{y}_t^\top Q \mathbf{y}_t$. We choose the latter because we do not want the cost to be tied to a particular (unknown) state representation.

[2]The one step delay in $\mathbf{y}_t$ is a standard assumption in controls which slightly simplifies the Kalman filtering expressions. Our results generalize to the setting where the history also contains the current observation $\mathbf{y}_t$.

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
