[Supplementary Material]

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

# A  Proof of Theorem 1

In this section we prove our meta theorem; we show how an upper bound $\|\widehat{P} - P_\star\| \leq f(\varepsilon)$ can be used to quantify the mismatch between the performance of the the nominal controller and the optimal controller. First, we upper bound $\|\widehat{K} - K_\star\|$ and offer a condition on this mismatch size so that $A_\star + B_\star\widehat{K}$ is a stable matrix. The next two optimization results are helpful in proving $\|\widehat{K} - K_\star\|$ is small.

**Lemma 1.** *Let $f_1, f_2$ be two $\mu$-strongly convex twice differentiable functions. Let $\mathbf{x}_1 = \arg\min_{\mathbf{x}} f_1(\mathbf{x})$ and $\mathbf{x}_2 = \arg\min_{\mathbf{x}} f_2(\mathbf{x})$. Suppose $\|\nabla f_1(\mathbf{x}_2)\| \leq \varepsilon$, then $\|\mathbf{x}_1 - \mathbf{x}_2\| \leq \frac{\varepsilon}{\mu}$.*

*Proof.* Taylor expanding $\nabla f_1$, we have:

$$\nabla f_1(\mathbf{x}_2) = \nabla f_1(\mathbf{x}_1) + \nabla^2 f_1(\tilde{\mathbf{x}})(\mathbf{x}_2 - \mathbf{x}_1) = \nabla^2 f_1(\tilde{\mathbf{x}})(\mathbf{x}_2 - \mathbf{x}_1) .$$

for $\tilde{\mathbf{x}} = t\mathbf{x}_1 + (1 - t)\mathbf{x}_2$ with some $t \in [0, 1]$. Therefore:

$$\mu\|\mathbf{x}_1 - \mathbf{x}_2\| \leq \|\nabla^2 f_1(\tilde{\mathbf{x}})(\mathbf{x}_2 - \mathbf{x}_1)\| = \|\nabla f_1(\mathbf{x}_2)\| \leq \varepsilon .$$

$\square$

**Lemma 2.** *Define $f_i(\mathbf{u}; \mathbf{x}) = \frac{1}{2}\mathbf{u}^\mathsf{T} R\mathbf{u} + \frac{1}{2}(A_i\mathbf{x} + B_i\mathbf{u})^\mathsf{T} P_i(A_i\mathbf{x} + B_i\mathbf{u})$ for $i = 1, 2$, with $R$, $P_1$, and $P_2$ positive definite matrices. Let $K_i$ be the unique matrix such that $\mathbf{u}_i := \arg\min_{\mathbf{u}} f_i(\mathbf{u}; \mathbf{x}) = K_i\mathbf{x}$ for any vector $\mathbf{x}$. Also, denote $\Gamma := 1 + \max\{\|A_1\|, \|B_1\|, \|P_1\|, \|K_1\|\}$. Suppose there exists $\varepsilon$ such that $0 \leq \varepsilon < 1$ and $\|A_1 - A_2\| \leq \varepsilon$, $\|B_1 - B_2\| \leq \varepsilon$, and $\|P_1 - P_2\| \leq \varepsilon$. Then, we have*

$$\|K_1 - K_2\| \leq \frac{7\varepsilon\Gamma^3}{\underline{\sigma}(R)} .$$

*Proof.* We first compute the gradient $\nabla f_i(\mathbf{u}; \mathbf{x})$ with respect to $\mathbf{u}$:

$$\nabla f_i(\mathbf{u}; \mathbf{x}) = (B_i^\mathsf{T} P_i B_i + R)\mathbf{u} + B_i^\mathsf{T} P_i A_i\mathbf{x} .$$

Now, we observe that:

$$\|B_1^\mathsf{T} P_1 B_1 - B_2^\mathsf{T} P_2 B_2\| \leq 7\Gamma^2\varepsilon \quad \text{and} \quad \|B_1^\mathsf{T} P_1 A_1 - B_2^\mathsf{T} P_2 A_2\| = 7\Gamma^2\varepsilon .$$

Hence, for any vector $\mathbf{x}$ with $\|\mathbf{x}\| \leq 1$, we have

$$\|\nabla f_1(\mathbf{u}; \mathbf{x}) - \nabla f_2(\mathbf{u}; \mathbf{x})\| \leq 7\Gamma^2\varepsilon(\|\mathbf{u}\| + 1) .$$

We can bound $\|\mathbf{u}_1\| \leq \|K_1\|\|\mathbf{x}\| \leq \|K_1\|$. Then, from Lemma 1 we obtain

$$\underline{\sigma}(R)\|(K_1 - K_2)\mathbf{x}\| = \underline{\sigma}(R)\|\mathbf{u}_1 - \mathbf{u}_2\| \leq 7\Gamma^3\varepsilon .$$

$\square$

Recall that $\Gamma_\star := 1 + \max\{\|A_\star\|, \|B_\star\|, \|P_\star\|, \|K_\star\|\}$. Now, we upper bound $\|\widehat{K} - K_\star\|$.

**Proposition 3.** *Let $\varepsilon > 0$ such that $\|\widehat{A} - A_\star\| \leq \varepsilon$ and $\|\widehat{B} - B_\star\| \leq \varepsilon$. Also, let $\|\widehat{P} - P_\star\| \leq f(\varepsilon)$ for some function $f$ such that $f(\varepsilon) \geq \varepsilon$. Then, under Assumption 1 we have*

$$\|\widehat{K} - K_\star\| \leq 7\Gamma_\star^3 f(\varepsilon). \tag{14}$$

*Let $\gamma$ be a real number such that $\rho(L_\star) < \gamma < 1$. Then, if $f(\varepsilon)$ is small enough so that the right hand side of (14) is smaller than $\frac{1-\gamma}{2\tau(L_\star, \gamma)}$, we have*

$$\tau\left(A_\star + B_\star K, \frac{1 + \gamma}{2}\right) \leq \tau(L_\star, \gamma).$$

*Proof.* By our assumptions $\|\widehat{A} - A_\star\|$, $\|\widehat{B} - B_\star\|$, and $\|\widehat{P} - P_\star\|$ are smaller than $f(\varepsilon)$, and $\underline{\sigma}(R) \geq 1$. Then, Lemma 2 ensures that

$$\|\widehat{K} - K_\star\| \leq 7\Gamma_\star^3 f(\varepsilon).$$

Finally, when $\varepsilon$ is small enough so that the right hand side of (14) is smaller or equal than $\frac{1-\gamma}{2\tau(A_\star + B_\star K, \gamma)}$, we can apply Lemma 5, presented in Section B, to guarantee that $\|(A_\star + B_\star\widehat{K})^k\| \leq \tau(A_\star + B_\star K_\star, \gamma)\left(\frac{1+\gamma}{2}\right)^k$ for all $k \geq 0$.

$\square$

In order to finish the proof of Theorem 1 we need to quantify the suboptimality gap $\widehat{J} - J_\star$ in terms of the controller mismatch $\widehat{K} - K_\star$. For a stable matrix $L$ and a symmetric matrix $M$, we let $\mathsf{dlyap}(L, M)$ denote the solution $X$ to the Lyapunov equation $L^\top X L - X + M = 0$. The following lemma offers a useful second order expansion of the average LQR cost.

**Lemma 3** (Lemma 12 of Fazel et al. [17]). *Let $K$ be an arbitrary static linear controller that stabilizes $(A_\star, B_\star)$. Denote $\Sigma(K) := \mathsf{dlyap}((A_\star + B_\star K)^\top, \sigma_w^2 I_n)$ the covariance matrix of the stationary distribution of the closed loop system $A_\star + B_\star K$. We have that:*

$$J(A_\star, B_\star, K) - J_\star = \mathrm{tr}(\Sigma(K)(K - K_\star)^\mathsf{T}(R + B_\star^\mathsf{T} P_\star B_\star)(K - K_\star)). \tag{15}$$

Now, we have the necessary ingredients to complete the proof of Theorem 1. Equation 15 implies:

$$J(A_\star, B_\star, K) - J_\star \leq \|\Sigma(K)\|\|R + B_\star^\mathsf{T} P_\star B_\star\|\|K - K_\star\|_F^2.$$

Proposition 3 states that $\widehat{K}$ stabilizes the system $(A_\star, B_\star)$ when the estimation error is small enough. More precisely, under the assumptions of Theorem 1, we have $\tau\left(A_\star + B_\star\widehat{K}, \frac{1+\gamma}{2}\right) \leq \tau(L_\star, \gamma)$. When $\widehat{L} = A_\star + B_\star\widehat{K}$ is a stable matrix we know that $\Sigma(K) = \sigma^2 \sum_{t\geq 0}(L^\top)^t L^t$. Then, by the triangle inequality we can bound

$$\|\Sigma(K)\| \leq \frac{\sigma_w^2 \tau(L_\star, \gamma)^2}{1 - \left(\frac{\gamma+1}{2}\right)^2} \leq \frac{4\sigma_w^2 \tau(L_\star, \gamma)^2}{1 - \gamma^2}.$$

Recalling that $\Gamma_\star := 1 + \max\{\|A_\star\|, \|B_\star\|, \|P_\star\|, \|K_\star\|\}$, we have $\|R + B_\star^\mathsf{T} P_\star B_\star\| \leq \Gamma^3$. Then,

$$J(K) - J_\star \leq 4\sigma_w^2 \Gamma^3 \frac{\tau(L_\star, \gamma)^2}{1 - \gamma^2}\|K - K_\star\|_F^2$$

$$\leq 4\sigma_w^2 \min\{n, d\}\Gamma^3 \frac{\tau(L_\star, \gamma)^2}{1 - \gamma^2}\|K - K_\star\|^2$$

$$\leq 200\sigma_w^2 d\Gamma^9 \frac{\tau(L_\star, \gamma)^2}{1 - \gamma^2} f(\varepsilon)^2,$$

where we used Proposition 3 and the assumption on $f(\varepsilon)$.

## B   Perturbation of the discrete algebraic Riccati equation

### B.1   Proof of Proposition 1

Given parameters $(A, B, Q)$ ($R$ is assumed fixed throughout; $Q$ is assumed positive semidefinite throughout) we denote by $F(X, A, B, Q)$ the matrix expression

$$F(X, A, B, Q) = X - A^\top X A + A^\top X B(R + B^\top X B)^{-1} B^\top X A - Q$$

$$= X - A^\top X \left(I + BR^{-1}B^\top X\right)^{-1} A - Q. \tag{16}$$

Then, solving the Riccati equation associated with $(A, B, Q)$ corresponds to finding the unique positive definite matrix $X$ such that $F(X, A, B, Q) = 0$. We denote by $P_\star$ the solution of the Riccati equation corresponding to the true system parameters $(A_\star, B_\star)$ and we denote by $\widehat{P}$ the solution associated with $(\widehat{A}, \widehat{B}, \widehat{Q})$. Our goal is to upper bound $\|\widehat{P} - P_\star\|$ in terms of $\varepsilon$, where $\varepsilon > 0$ such that $\|\widehat{A} - A_\star\| \leq \varepsilon$, $\|\widehat{B} - B_\star\| \leq \varepsilon$, and $\|\widehat{Q} - Q_\star\| \leq \varepsilon$.

We denote $\Delta_P = \widehat{P} - P_\star$. The proof strategy goes as follows. Given the identities $F(P_\star, A_\star, B_\star, Q_\star) = 0$ and $F(\widehat{P}, \widehat{A}, \widehat{B}, \widehat{Q}) = 0$ we construct an operator $\Phi$ such that $\Delta_P$ is its unique fixed point. Then, we show that the fixed point of $\Phi$ must have small norm when $\varepsilon$ is sufficiently small.

We denote $S_\star = B_\star R^{-1} B_\star^\top$ and $\widehat{S} = \widehat{B} R^{-1}\widehat{B}^\top$. Also, recall that $L_\star = A_\star + B_\star K_\star$. For any matrix $X$ such that $I + S_\star(P_\star + X)$ is invertible we have

$$F(P_\star + X, A_\star, B_\star, Q_\star) = X - L_\star^\top X L_\star + L_\star^\top X \left[I + S_\star(P_\star + X)\right]^{-1} S_\star X L_\star. \tag{17}$$

To check this identity one needs to add $F(P_\star, A_\star, B_\star, Q_\star)$, which is equal to zero, to the right hand side of (17) and use the identity $(I + B_\star R^{-1} B_\star^\top P_\star)^{-1} A_\star = A_\star + B_\star K_\star$. This last identity can be checked by recalling $K_\star = -(R + B_\star^\top P_\star B_\star)^{-1} B_\star^\top P_\star A_\star$ and using the matrix inversion formulat.

To write (17) more compactly we define the following two matrix operators

$$\mathcal{T}(X) = X - L_\star^\top X L_\star \text{ and } \mathcal{H}(X) = L_\star^\top X \left(I + S_\star(P_\star + X)\right)^{-1} S_\star X L_\star.$$

Then, Equation 17 becomes $F(P_\star + X, A_\star, B_\star, Q_\star) = \mathcal{T}(X) + \mathcal{H}(X)$. Since Equation 17 is satisfied by any matrix $X$ with $I + S_\star(P_\star + X)$ invertible, the matrix equation

$$F(P_\star + X, A_\star, B_\star, Q_\star) - F(P_\star + X, \widehat{A}, \widehat{B}, \widehat{Q}) = \mathcal{T}(X) + \mathcal{H}(X) \tag{18}$$

has a unique symmetric solution $X$ such that $P_\star + X \succeq 0$. That solution is $X = \Delta_P$ because any solution of (18) must satisfy $F(P_\star + X, \widehat{A}, \widehat{B}, \widehat{Q}) = 0$.

The linear map $\mathcal{T} \colon X \mapsto X - L_\star^\top X L_\star$ has eigenvalues equal to $1 - \lambda_i \lambda_j$, where $\lambda_i$ and $\lambda_j$ are eigenvalues of the closed loop matrix $L_\star$. Since $L_\star$ is a stable matrix, the linear map $\mathcal{T}$ must be invertible. Now, we define the operator

$$\Phi(X) = \mathcal{T}^{-1}\left(F(P_\star + X, A_\star, B_\star, Q_\star) - F(P_\star + X, \widehat{A}, \widehat{B}, \widehat{Q}) - \mathcal{H}(X)\right).$$

Then, solving for $X$ in Equation 18 is equivalent to finding $X$ satisfying $P_\star + X \succeq 0$ such that $X = \Phi(X)$. Hence, $\Phi$ has a unique symmetric fixed point $X$ such that $P_\star + X \succeq 0$ and that is $X = \Delta_P$. Now, we consider the set

$$\mathcal{S}_\nu := \left\{X \colon \|X\| \leq \nu, X = X^\top, P_\star + X \succeq 0\right\}$$

and we show that for an appropriately chosen $\nu$ the operator $\Phi$ maps $\mathcal{S}_\nu$ into itself and is also a contraction over the set $\mathcal{S}_\nu$. If we show these two properties, $\Phi$ is guaranteed to have a fixed point in the set $\mathcal{S}_\nu$. However, since $\Delta_P$ is the only possible fixed point of $\Phi$ in a set $\mathcal{S}_\nu$ we find $\|\Delta_P\| \leq \nu$.

We denote $\Delta_A = \widehat{A} - A_\star$, $\Delta_B = \widehat{B} - B_\star$, $\Delta_Q = \widehat{Q} - Q_\star$, and $\Delta_S = \widehat{S} - S_\star$. By assumption we have $\|\Delta_A\| \leq \varepsilon$, $\|\Delta_B\| \leq \varepsilon$, $\|\Delta_Q\| \leq \varepsilon$. Then, $\|\Delta_S\| \leq 3\|B_\star\|\|R^{-1}\|\varepsilon$ because $\varepsilon \leq \|B_\star\|$.

**Lemma 4.** *Suppose the matrices $X$, $X_1$, $X_2$ belong to $\mathcal{S}_\nu$, with $\nu \leq \min\{1, \|S_\star\|^{-1}\}$. Furthermore, we assume that $\|\Delta_A\| \leq \varepsilon$, $\|\Delta_B\| \leq \varepsilon$, and $\|\Delta_Q\| \leq \varepsilon$ with $\varepsilon \leq \min\{1, \|B_\star\|\}$. Finally, let $\underline{\sigma}(P_\star) \geq 1$. Then*

$$\|\Phi(X)\| \leq 3\frac{\tau(L_\star, \gamma)^2}{1 - \gamma^2}\left[\|L_\star\|^2\|S_\star\|\nu^2 + \varepsilon\|A_\star\|_+^2\|P_\star\|_+^2\|B_\star\|_+\|R^{-1}\|_+\right],$$

$$\|\Phi(X_1) - \Phi(X_2)\| \leq 32\frac{\tau(L_\star, \gamma)^2}{1 - \gamma^2}\left[\|L_\star\|^2\|S_\star\|\nu + \varepsilon\|A_\star\|_+^2\|P_\star\|_+^3\|B_\star\|_+^3\|R^{-1}\|_+^2\right]\|X_1 - X_2\|.$$

The proof of this lemma is defered to Appendix E. Now, we choose

$$\nu = 6\,\varepsilon\,\frac{\tau(L_\star, \gamma)^2}{1 - \gamma^2}\|A_\star\|_+^2\|P_\star\|_+^2\|B_\star\|_+\|R^{-1}\|_+. \tag{19}$$

Since $\varepsilon$ is assumed to be small enough, we know

$$\nu \leq \min\left\{\frac{1 - \gamma^2}{128\tau(L_\star, \gamma)^2\|L_\star\|^2\|S_\star\|}, \|S_\star\|^{-1}, \frac{1}{2}\right\}.$$

Then, the operator $\Phi$ satisfies $\|\Phi(X_1) - \Phi(X_2)\| \leq \frac{1}{2}\|X_1 - X_2\|$ for all $X_1$ and $X_2$ in $\mathcal{S}_\nu$. Moreover, we have $\|\Phi(X)\| \leq \nu$ for all $X \in \mathcal{S}_\nu$. Since $\nu \leq \underline{\sigma}(P_\star)$, we know that $P_\star + \Phi(X) \succeq 0$

Therefore, $\Phi$ maps $\mathcal{S}_\nu$ into itself and is a contraction over $\mathcal{S}_\nu$. Hence, $\Phi$ has a fixed point in $\mathcal{S}_\nu$ since $\mathcal{S}_\nu$ is a closed set. However, we already argued that the unique fixed point of $\Phi$ is $\Delta_P$. Therefore, $\Delta_P \in \mathcal{S}_\nu$ and $\|\Delta_P\| \leq \nu$. Proposition 1 is now proven.

## B.2 Proof of Proposition 2

Since both noisy and noiseless LQR have the same associated Riccati equation and the same optimal controller, we can focus on the noiseless case in this section. Namely, noiseless LQR takes the form

$$\min_{\mathbf{u}} \sum_{t=0}^{\infty} \mathbf{x}_t^{\top} Q \mathbf{x}_t + \mathbf{u}_t^{\top} R \mathbf{u}_t \text{ , where } \mathbf{x}_{t+1} = A_{\star} \mathbf{x}_t + B_{\star} \mathbf{u}_t,$$

for a given initial state $\mathbf{x}_0$. Then, we know that the cost achieved by the optimal controller when the system is initialized at $\mathbf{x}_0$ is equal to $\mathbf{x}_0^{\top} P_{\star} \mathbf{x}_0$.

We denote by $J(A, B, \mathbf{x}_0, \{\mathbf{u}_t\}_{t \geq 0})$ the cost achieved on a linear system $(A, B)$ initialized at $\mathbf{x}_0$ by the input sequence $\{\mathbf{u}_t\}_{t \geq 0}$. When the input sequence is given by a time invariant linear gain matrix $K$ we slightly abuse notation and denote the cost by $J(A, B, \mathbf{x}_0, K)$. In this case, $J(A, B, \mathbf{x}_0, K) = \mathbf{x}_0^{\top} P \mathbf{x}_0$, where $P$ is the solution to the associated Riccati equation.

Now, let $\mathbf{x}_0$ be an arbitrary unit state vector in $\mathbb{R}^n$. Then,

$$\mathbf{x}_0^{\top} \widehat{P} \mathbf{x}_0 - \mathbf{x}_0^{\top} P_{\star} \mathbf{x}_0 = J(\widehat{A}, \widehat{B}, \mathbf{x}_0, \widehat{K}) - J(A_{\star}, B_{\star}, \mathbf{x}_0, K_{\star})$$
$$\leq J(\widehat{A}, \widehat{B}, \mathbf{x}_0, \{\widehat{\mathbf{u}}_t\}_{t \geq 0}) - J(A_{\star}, B_{\star}, \mathbf{x}_0, K_{\star})$$

for any sequence of inputs $\{\widehat{\mathbf{u}}_t\}_{t \geq 0}$. We denote by $\widehat{\mathbf{x}}_t$ the states produced by $\widehat{\mathbf{u}}_t$ on the system $(\widehat{A}, \widehat{B})$ and by $\mathbf{x}_t$ and $\mathbf{u}_t$ the states and actions obtained on the system $(A_{\star}, B_{\star})$ when the optimal controller $\mathbf{u}_t = K_{\star} \mathbf{x}_t$ is used. To prove Proposition 2 we choose a sequence of actions $\{\widehat{\mathbf{u}}_t\}_{t \geq 0}$ such that $J(\widehat{A}, \widehat{B}, \mathbf{x}_0, \{\widehat{\mathbf{u}}_t\}_{t \geq 0}) \approx J(A_{\star}, B_{\star}, \mathbf{x}_0, K_{\star})$.

For any sequence of inputs $\{\widehat{\mathbf{u}}_t\}_{t \geq 0}$ such that the series defining the cost $J(\widehat{A}, \widehat{B}, \mathbf{x}_0, \{\widehat{\mathbf{u}}_t\}_{t \geq 0})$ is absolutely convergent, we can write

$$J(\widehat{A}, \widehat{B}, \mathbf{x}_0, \widehat{K}) - J(A_{\star}, B_{\star}, \mathbf{x}_0, K_{\star}) = \sum_{j=0}^{\infty} \sum_{i=0}^{\ell-1} \left[ \widehat{\mathbf{x}}_{\ell j + i}^{\top} Q \widehat{\mathbf{x}}_{\ell j + i} - \mathbf{x}_{\ell j + i}^{\top} Q \mathbf{x}_{\ell j + i} \right] \quad (20)$$
$$+ \sum_{j=0}^{\infty} \sum_{i=0}^{\ell-1} \left[ \widehat{\mathbf{u}}_{\ell j + i}^{\top} R \widehat{\mathbf{u}}_{\ell j + i} - \mathbf{u}_{\ell j + i}^{\top} R \mathbf{u}_{\ell j + i} \right].$$

Then, the key idea is to choose a sequence of inputs $\{\widehat{\mathbf{u}}_t\}_{t \geq 0}$ such that the system $(\widehat{A}, \widehat{B})$ tracks the system $(A_{\star}, B_{\star}, K_{\star})$, i.e., $\widehat{\mathbf{x}}_{\ell j} = \mathbf{x}_{\ell j}$ for any $j \geq 0$ ( $\widehat{\mathbf{x}}_0 = \mathbf{x}_0$ because both systems are initialized at the same state). This can be done because $(\widehat{A}, \widehat{B})$ is $(\ell, \tau/2)$-controllable when $(A_{\star}, B_{\star})$ is $(\ell, \tau)$-controllable and the estimation error is sufficiently small, as shown in Lemma 6. First, we present a result that quantifies the effect of matrix perturbations on powers of matrices.

**Lemma 5.** *Let $M$ be an arbitrary matrix in $\mathbb{R}^{n \times n}$ and let $\rho \geq \rho(M)$. Then, for all $k \geq 1$ and real matrices $\Delta$ of appropriate dimensions we have*

$$\|(M + \Delta)^k\| \leq \tau(M, \rho)(\tau(M, \rho)\|\Delta\| + \rho)^k,$$
$$\|(M + \Delta)^k - M^k\| \leq k \, \tau(M, \rho)^2 (\tau(M, \rho)\|\Delta\| + \rho)^{k-1} \|\Delta\|.$$

*Recall that $\tau(M, \rho)$ is defined in Equation 3.*

The proof is deferred to Appendix C. Lemma 5 quantifies the effect of a perturbation $\Delta$, applied to a matrix $M$ on the spectral radius of $M + \Delta$. We are interested in quantifying the sizes of these perturbations for all $k = 1, 2, \ldots, \ell$. Depending on $\|\Delta\|$, $M$, and $\rho$ the sum $\tau(M, \rho)\|\Delta\| + \rho$ can either be greater than one or smaller than one. For notational simplicity, in the rest of the proof we denote $\beta = \max\{1, \varepsilon\tau(A_{\star}, \rho) + \rho\}$. Then, we have $\|(A_{\star} + \Delta)^k\| \leq \tau(A_{\star}, \rho)\beta^{\ell-1}$ and $\|(A_{\star} + \Delta)^k - A_{\star}^k\| \leq \ell\tau(A_{\star}, \rho)^2\beta^{\ell-1}\varepsilon$ for all $k \leq \ell - 1$ and all real matrices $\Delta$ with $\|\Delta\| \leq \varepsilon$.

We denote $C_{\ell} = \begin{bmatrix} B_{\star} & A_{\star}B_{\star} & \ldots & A_{\star}^{\ell-1}B_{\star} \end{bmatrix}$ and $\widehat{C}_{\ell} = \begin{bmatrix} \widehat{B} & \widehat{A}\widehat{B} & \ldots & \widehat{A}^{\ell-1}\widehat{B} \end{bmatrix}$. Before presenting the next result we recall that for any block matrix $M$ with blocks $M_{i,j}$ we have $\|M\|^2 \leq \sum_{i,j} \|M_{i,j}\|^2$. The next lemma gives us control over the smallest positive singular value of the controllability matrix $\widehat{C}_{\ell}$ in terms of the corresponding value for $C_{\ell}$.

**Lemma 6.** *Suppose the linear $(A_\star, B_\star)$ is $(\ell, \nu)$-controllable and let $\rho$ be a real number such that $\rho \geq \rho(A_\star)$. Then, if $\|\widehat{A} - A_\star\| \leq \varepsilon$ and $\|\widehat{B} - B_\star\| \leq \varepsilon$, we have*

$$\underline{\sigma}(\widehat{C}_\ell) \geq \tau - 3\varepsilon\ell^{\frac{3}{2}}\tau(A_\star, \rho)^2 \max\{1, \tau(A_\star, \rho)\|\Delta\| + \rho\}^{\ell-1} (\|B_\star\| + 1).$$

The proof is deferred to Appendix D. Lemma 6 tells us that by the assumption made in Proposition 2 on $\varepsilon$, we have $\underline{\sigma}(\widehat{C}_\ell) \geq \frac{\tau_\ell}{2}$. Hence, we know that for any $\mathbf{x}_0 \in \mathbb{R}^n$ and $\mathbf{u}_0, \mathbf{u}_1, \ldots, \mathbf{u}_{\ell-1} \in \mathbb{R}^d$, there exist $\widehat{\mathbf{u}}_0, \widehat{\mathbf{u}}_1, \ldots, \widehat{\mathbf{u}}_{\ell-1} \in \mathbb{R}^d$ such that

$$A_\star^\ell \mathbf{x}_0 + \sum_{i=0}^{\ell-1} A_\star^i B_\star \mathbf{u}_{\ell-1-i} = \widehat{A}^\ell \mathbf{x}_0 + \sum_{i=0}^{\ell-1} \widehat{A}^i \widehat{B} \widehat{\mathbf{u}}_{\ell-1-i} \qquad (21)$$

because the system $(\widehat{A}, \widehat{B})$ is controllable. This equation implies that $\widehat{\mathbf{x}}_\ell = \mathbf{x}_\ell$.

We denote the concatenation of $\mathbf{u}_i$, for $i$ from 0 to $\ell - 1$ by $\mathbf{u}^{(\ell)}$. We define $\widehat{\mathbf{u}}^{(\ell)}$ analogously. Therefore, Equation (21) can be rewritten as

$$\left(A_\star^\ell - \widehat{A}^\ell\right) \mathbf{x}_0 + \left(\mathcal{C}_\ell - \widehat{\mathcal{C}}_\ell\right) \mathbf{u}^{(\ell)} = \widehat{\mathcal{C}}_\ell(\widehat{\mathbf{u}}^{(\ell)} - \mathbf{u}^{(\ell)}). \qquad (22)$$

Recall that $\beta = \max\{1, \tau(A_\star, \rho)\|\Delta\| + \rho\}$. Combining Lemma 5 and the upper bound on operator norms of block matrices we find $\|\widehat{\mathcal{C}}_\ell - \mathcal{C}_\ell\| \leq \varepsilon\ell^{\frac{3}{2}}\tau(A_\star, \rho)^2\beta^{\ell-1} (\|B_\star\| + 1)$.

We are free to choose $\widehat{\mathbf{u}}^{(\ell)}$ anyway we wish as long as Equation (22) is true. Therefore, we can choose $\widehat{\mathbf{u}}^{(\ell)}$ such that $\widehat{\mathbf{u}}^{(\ell)} - \mathbf{u}^{(\ell)}$ is perpendicular to the nullspace of $\widehat{\mathcal{C}}_\ell$. Then,

$$\frac{\tau_\ell}{2}\|\widehat{\mathbf{u}}^{(\ell)} - \mathbf{u}^{(\ell)}\| \leq \|\widehat{\mathcal{C}}_\ell(\widehat{\mathbf{u}}^{(\ell)} - \mathbf{u}^{(\ell)})\| \leq \varepsilon\ell\tau(A_\star, \rho)^2\beta^{\ell-1}\|\mathbf{x}_0\| + \|\widehat{\mathcal{C}}_\ell - \mathcal{C}_\ell\|\|\mathbf{u}^{(\ell)}\|$$

$$\leq \varepsilon\ell\tau(A_\star, \rho)^2\beta^{\ell-1}\|\mathbf{x}_0\| + \varepsilon\ell^{\frac{3}{2}}\tau(A_\star, \rho)^2\beta^{\ell-1} (\|B_\star\| + 1) \|\mathbf{u}^{(\ell)}\|.$$

Hence,

$$\|\widehat{\mathbf{u}}^{(\ell)} - \mathbf{u}^{(\ell)}\| \leq \frac{2\varepsilon\ell^{\frac{3}{2}}}{\tau_\ell}\tau(A_\star, \rho)^2\beta^{\ell-1}(\|B_\star\| + 1)\left(\|\mathbf{x}_0\| + \|\mathbf{u}^{(\ell)}\|\right)$$

$$=: \eta\left(\|\mathbf{x}_0\| + \|\mathbf{u}^{(\ell)}\|\right). \qquad (23)$$

Let us consider the block Toeplitz matrix

$$\mathcal{T}_\ell = \begin{bmatrix} 0 & 0 & 0 & \ldots & 0 \\ B_\star & 0 & 0 & \cdots & 0 \\ A_\star B_\star & B_\star & 0 & \cdots & 0 \\ \vdots & \cdots & \cdots & \cdots & \cdots \\ A_\star^{\ell-2} B_\star & A_\star^{\ell-3} B_\star & \cdots & B_\star & 0 \end{bmatrix}.$$

From Lemma 5 and the upper bound on operator norms of block matrices we have $\|\mathcal{T}_\ell - \widehat{\mathcal{T}}_\ell\| \leq \varepsilon\ell^2\tau(A_\star, \rho)^2\beta^{\ell-2}(\|B_\star\| + 1)$. Let $\mathbf{x}^{(\ell)}$ be the concatenation of the vectors $\mathbf{x}_0, \mathbf{x}_1, \ldots, \mathbf{x}_{\ell-1}$. Then,

$$\mathbf{x}^{(\ell)} = \mathcal{T}_\ell \mathbf{u}^{(\ell)} + \begin{bmatrix} I_n \\ A_\star \\ \vdots \\ A^{\ell-1} \end{bmatrix} \mathbf{x}_0.$$

Hence,

$$\|\mathbf{x}^{(\ell)} - \widehat{\mathbf{x}}^{(\ell)}\| \leq \|\mathcal{T}_\ell \mathbf{u}^{(\ell)} - \widehat{\mathcal{T}}_\ell \widehat{\mathbf{u}}^{(\ell)}\| + \varepsilon\ell^{\frac{3}{2}}\tau(A_\star, \rho)^2\beta^{\ell-2}\|\mathbf{x}_0\|$$

$$\leq \|\mathcal{T}_\ell \mathbf{u}^{(\ell)} - \widehat{\mathcal{T}}_\ell \mathbf{u}^{(\ell)}\| + \|\widehat{\mathcal{T}}_\ell \mathbf{u}^{(\ell)} - \widehat{\mathcal{T}}_\ell \widehat{\mathbf{u}}^{(\ell)}\| + \varepsilon\ell^{\frac{3}{2}}\tau(A_\star, \rho)^2\beta^{\ell-2}\|\mathbf{x}_0\|$$

$$\leq \|\mathcal{T}_\ell - \widehat{\mathcal{T}}_\ell\|\|\mathbf{u}^{(\ell)}\| + \|\widehat{\mathcal{T}}_\ell\|\|\mathbf{u}^{(\ell)} - \widehat{\mathbf{u}}^{(\ell)}\| + \varepsilon\ell^{\frac{3}{2}}\tau(A_\star, \rho)^2\beta^{\ell-2}\|\mathbf{x}_0\|$$

$$\leq \varepsilon\ell^2\tau(A_\star, \rho)^2\beta^{\ell-2}(\|B_\star\| + 1)\|\mathbf{u}^{(\ell)}\| + \varepsilon\ell^{\frac{3}{2}}\tau(A_\star, \rho)^2\beta^{\ell-2}\|\mathbf{x}_0\|$$

$$\quad + \ell\tau(A_\star, \rho)\beta^{\ell-2}(\|B_\star\| + 1)\|\mathbf{u}^{(\ell)} - \widehat{\mathbf{u}}^{(\ell)}\|$$

$$\leq 2\varepsilon\ell^{\frac{5}{2}}\tau(A_\star, \rho)^3\beta^{2(\ell-1)}(1 + \tau^{-1})(\|B_\star\| + 1)^2 \left[\|\mathbf{x}_0\| + \|\mathbf{u}^{(\ell)}\|\right]$$

$$=: \mu\left[\|\mathbf{u}^{(\ell)}\| + \|\mathbf{x}_0\|\right]. \qquad (24)$$

In Equations (23) and (24) we proved that the inputs and states of the system $(\widehat{A}, \widehat{B})$ are close to the inputs and states of the system $(A_\star, B_\star)$ from time 0 to $\ell$. Since the inputs to the system $(\widehat{A}, \widehat{B})$ satisfy Equation (22), we know that $\widehat{\mathbf{x}}_{\ell j} = \mathbf{x}_{\ell j}$ for all $j$. We can repeat the same argument as above, with $\mathbf{x}_{\ell j}$ taking the place of $\mathbf{x}_0$, to show that the inputs and states of the two systems are close to each other from time $\ell j$ to $\ell(j+1)$. Let us denote by $\mathbf{x}_j^{(\ell)}$ the concatenation of the vectors $\mathbf{x}_{\ell j}, \mathbf{x}_{\ell j+1}, \ldots,$ $\mathbf{x}_{\ell j+\ell-1}$ and let $\mathbf{u}_j^{(\ell)}$ be defined analogously. Then,

$$\|\widehat{\mathbf{u}}_j^{(\ell)} - \mathbf{u}_j^{(\ell)}\| \le \eta \left[ \|\mathbf{u}_j^{(\ell)}\| + \|\mathbf{x}_{\ell j}\| \right], \quad \text{and} \quad \|\widehat{\mathbf{x}}_j^{(\ell)} - \mathbf{x}_j^{(\ell)}\| \le \mu \left[ \|\mathbf{u}_j^{(\ell)}\| + \|\mathbf{x}_{\ell j}\| \right]. \quad (25)$$

Now, we note that

$$\begin{aligned}
\mathbf{x}_0^\top \widehat{P} \mathbf{x}_0 - \mathbf{x}_0^\top P_\star \mathbf{x}_0 &\le \sum_{j=0}^{\infty} \sum_{i=0}^{\ell-1} \left[ \widehat{\mathbf{x}}_{\ell j+i}^\top Q \widehat{\mathbf{x}}_{\ell j+i} - \mathbf{x}_{\ell j+i}^\top Q \mathbf{x}_{\ell j+i} \right] \\
&\quad + \sum_{j=0}^{\infty} \sum_{i=0}^{\ell-1} \left[ \widehat{\mathbf{u}}_{\ell j+i}^\top R \widehat{\mathbf{u}}_{\ell j+i} - \mathbf{u}_{\ell j+i}^\top R \mathbf{u}_{\ell j+i} \right] \\
&\le \sum_{j=0}^{\infty} 2\|Q\| \|\mathbf{x}_j^{(\ell)}\| \|\mathbf{x}_j^{(\ell)} - \widehat{\mathbf{x}}_j^{(\ell)}\| + \|Q\| \|\mathbf{x}_j^{(\ell)} - \widehat{\mathbf{x}}_j^{(\ell)}\|^2 \\
&\quad + \sum_{j=0}^{\infty} 2\|R\| \|\mathbf{u}_j^{(\ell)}\| \|\mathbf{u}_j^{(\ell)} - \widehat{\mathbf{u}}_j^{(\ell)}\| + \|R\| \|\mathbf{u}_j^{(\ell)} - \widehat{\mathbf{u}}_j^{(\ell)}\|^2.
\end{aligned}$$

Now, we use the upper bounds from (25). We always have $\eta \le \mu$. Since Proposition 2 assumes $\varepsilon$ is small enough, we also have $\mu \le 1$. Using these upper bounds, we find

$$\begin{aligned}
\widehat{J} - J_\star &\le \mu \sum_{j=0}^{\infty} 2\|Q\| \|\mathbf{x}_j^{(\ell)}\| \left[ \|\mathbf{u}_j^{(\ell)}\| + \|\mathbf{x}_{\ell j}\| \right] + \|Q\| \left[ \|\mathbf{u}_j^{(\ell)}\| + \|\mathbf{x}_{\ell j}\| \right]^2 \\
&\quad + \mu \sum_{j=0}^{\infty} 2\|R\| \|\mathbf{u}_j^{(\ell)}\| \left[ \|\mathbf{u}_j^{(\ell)}\| + \|\mathbf{x}_{\ell j}\| \right] + \|R\| \left[ \|\mathbf{u}_j^{(\ell)}\| + \|\mathbf{x}_{\ell j}\| \right]^2.
\end{aligned}$$

Then, we get $\widehat{J} - J_\star \le 8\mu \max\{\|Q\|, \|R\|\} \sum_{j=0}^{\infty} \|\mathbf{x}_j^{(\ell)}\|^2 + \|\mathbf{u}_j^{(\ell)}\|^2$ after using the inequalities $(a+b)^2 \le 2(a^2+b^2)$ and $2ab \le a^2 + b^2$. Now, As long as $\|\mathbf{x}_0\| \le 1$ we have

$$\min\{\underline{\sigma}(Q), \underline{\sigma}(R)\} \sum_{j=0}^{\infty} \|\mathbf{x}_j^{(\ell)}\|^2 + \|\mathbf{u}_j^{(\ell)}\|^2 \le \sum_{t=0}^{\infty} \mathbf{x}_t^\top Q \mathbf{x}_t + \mathbf{u}_t^\top R \mathbf{u}_t = \mathbf{x}_0^\top P_\star \mathbf{x}_0 \le \|P_\star\|. \quad (26)$$

Since the initial state is an arbitrary unit norm vector, our upper bound on $\mathbf{x}_0^\top (\widehat{P} - P_\star) \mathbf{x}_0$ becomes

$$\lambda_{\max} \left( \widehat{P} - P_\star \right) \le 16 \varepsilon \ell^{\frac{5}{2}} \tau(A_\star, \rho)^3 \beta^{2(\ell-1)} (1 + \nu^{-1})(1 + \|B_\star\|)^2 \|P_\star\| \frac{\max\{\|Q\|, \|R\|\}}{\min\{\underline{\sigma}(Q), \underline{\sigma}(R)\}}. \quad (27)$$

Now, we can reverse the roles of $(\widehat{A}, \widehat{B})$ and $(A_\star, B_\star)$ and repeat the same argument and obtain an upper bound on $\lambda_{\max} \left( P_\star - \widehat{P} \right)$ analogous to Equation (27), but which has $\|P_\star\|$ replaced by $\|\widehat{P}\|$ on the right hand side. However, (27) implies that $\|\widehat{P}\| \le \|P_\star\| + 1 \le 2\|P_\star\|$ because we assumed that $\varepsilon$ is small enough such that the right hand side of (27) is less than one, and because $P_\star \succeq I_n$. The conclusion follows.

We note that the proof can be easily extended to the case when the cost matrix $Q$ is also being perturbed. Moreover, the only essential step in the argument where we used $Q \succ 0$ is (26). The goal of (26) is to upper bound $\sum_{j=0}^{\infty} \|\mathbf{x}_j^{(\ell)}\|^2 + \|\mathbf{u}_j^{(\ell)}\|^2$. This quantity can be upper bounded even when $Q$ is not positive definite, but the system $(Q^{1/2}, A)$ is observable; which is a necessary requirement for LQR on the parameters $(A, B, Q, R)$ to stabilize the system $(A, B)$.

# C  Proof of Lemma 5

This proof is a simple modification of Lemma D.1 in [11]. We replicate the argument here for completeness.

Fix an integer $k \geq 1$. Consider the expansion of $(M + \Delta)^k$ into $2^k$ terms. Label all these terms as $T_{i,j}$ for $i = 0, ..., k$ and $j = 1, ..., \binom{k}{i}$ where $i$ denotes the degree of $\Delta$ in the term (hence there are $\binom{k}{i}$ terms with a degree of $i$ for $\Delta$). Using the fact that $\|M^k\| \leq \tau(M, \rho)\rho^k$ for all $k \geq 0$, we can bound $\|T_{i,j}\| \leq \tau(M, \rho)^{i+1}\rho^{k-i}\|\Delta\|^i$. Hence by triangle inequality:

$$
\begin{aligned}
\|(M + \Delta)^k\| &\leq \sum_{i=0}^{k} \sum_{j} \|T_{i,j}\| \\
&\leq \sum_{i=0}^{k} \binom{k}{i} \tau(M, \rho)^{i+1} \rho^{k-i} \|\Delta\|^i \\
&= \tau(M, \rho) \sum_{i=0}^{k} \binom{k}{i} (\tau(M, \rho)\|\Delta\|)^i \rho^{k-i} \\
&= \tau(M, \rho)(\tau(M, \rho)\|\Delta\| + \rho)^k.
\end{aligned}
$$

To prove the first part of the lemma we follow the same argument. We find

$$
\begin{aligned}
\|(M + \Delta)^k - M^k\| &\leq \sum_{i=1}^{k} \sum_{j} \|T_{i,j}\| \\
&\leq \sum_{i=1}^{k} \binom{k}{i} \tau(M, \rho)^{i+1} \rho^{k-i} \|\Delta\|^i \\
&= \tau(M, \rho) \sum_{i=1}^{k} \binom{k}{i} (\tau(M, \rho)\|\Delta\|)^i \rho^{k-i} \\
&= \tau(M, \rho) \left[ (\tau(M, \rho)\|\Delta\| + \rho)^k - \rho^k \right] \\
&\leq k C_M^2 (\tau(M, \rho)\|\Delta\| + \rho)^{k-1} \|\Delta\|,
\end{aligned}
$$

where the last inequality follows from the mean value theorem applied to the function $z \mapsto z^k$.

# D  Proof of Lemma 6

We can write

$$
\underline{\sigma} \left( \begin{bmatrix} \widehat{B} & \widehat{A}\widehat{B} & \dots & \widehat{A}^{\ell-1}\widehat{B} \end{bmatrix} \right) = \min_{v \in \mathcal{S}^{n-1}} \left\| v^\top \begin{bmatrix} \widehat{B} & \widehat{A}\widehat{B} & \dots & \widehat{A}^{\ell-1}\widehat{B} \end{bmatrix} \right\|.
$$

Fix an arbitrary unit vector $v$ in $\mathbb{R}^d$. Then,

$$
\begin{aligned}
&\left\| v^\top \begin{bmatrix} B_\star & A_\star B_\star & \dots & A_\star^{\ell-1} B_\star \end{bmatrix} - v^\top \begin{bmatrix} \widehat{B} & \widehat{A}\widehat{B} & \dots & \widehat{A}^{\ell-1}\widehat{B} \end{bmatrix} \right\| \\
&\leq \left\| v^\top \begin{bmatrix} B_\star & A_\star B_\star & \dots & A_\star^{\ell-1} B_\star \end{bmatrix} - v^\top \begin{bmatrix} B_\star & \widehat{A}B_\star & \dots & \widehat{A}^{\ell-1} B_\star \end{bmatrix} \right\| \\
&\quad + \left\| v^\top \begin{bmatrix} B_\star & \widehat{A}B_\star & \dots & \widehat{A}^{\ell-1} B_\star \end{bmatrix} - v^\top \begin{bmatrix} \widehat{B} & \widehat{A}\widehat{B} & \dots & \widehat{A}^{\ell-1}\widehat{B} \end{bmatrix} \right\| \\
&\leq \varepsilon \ell^{\frac{3}{2}} \tau(A_\star, \rho)^2 \beta^{\ell-1} \|B_\star\| + \varepsilon \sqrt{\ell} \tau(A_\star, \rho, \beta)^{\ell-1} \\
&\leq \varepsilon \ell^{\frac{3}{2}} \tau(A_\star, \rho)^2 \beta^{\ell-1} (\|B_\star\| + 1).
\end{aligned}
$$

We used $\ell \geq 1$, $\tau(A_\star, \rho) \geq 1$, Lemma 5, and the upper bound $\|M\|^2 \leq \sum_{i,j} \|M_{i,j}\|^2$ on the operator norm of a block matrix. The conclusion follows by the triangle inequality.

# E   Proof of Lemma 4

We wish to upper bound $\|\Phi(X)\|$ and $\|\Phi(X_1) - \Phi(X_2)\|$ for $X$, $X_1$, and $X_2$ in $\mathcal{S}_\nu$. First, we upper bound the operator norm of the linear operator $\mathcal{T}^{-1}$, the inverse of $\mathcal{T}\colon X \mapsto X - L_\star^\top X L_\star$. Since $L_\star$ is a stable matrix, the linear map $\mathcal{T}$ must be invertible. Moreover, when $L_\star$ is stable and $X - L_\star^\top X L_\star = M$ for some matrix $M$, we know that $X = \sum_{k=0}^{\infty} (L_\star^k)^\top M L_\star^k$. Therefore, by the triangle inequality, the operator norm of $\mathcal{T}^{-1}$ can be upper bounded by $\|\mathcal{T}^{-1}\| \leq \frac{\tau(L_\star, \rho)^2}{1 - \rho^2}$. Before we proceed with the rest of the proof we present a technical lemma which will be used several times.

**Lemma 7.** *Let $M$ and $N$ be two positive semidefinite matrices of the same dimension. Then* $\|N(I + MN)^{-1}\| \leq \|N\|$.

*Proof.* We assume that $M$ and $N$ are invertible. If they are not, we can work with the matrices $M + \nu I$ and $N + \nu I$ and take the limit of $\nu$ going to zero. Then, we have $N(I + MN)^{-1} = NN^{-1}(N^{-1} + M)^{-1} = (N^{-1} + M)^{-1} \preceq N$. $\qquad\square$

Next, recall that $\mathcal{H}(X) = L_\star^\top X \left(I + S_\star(P_\star + X)\right)^{-1} S_\star X L_\star$. Then, Lemma 7 yields

$$\|\mathcal{H}(X)\| \leq \|L_\star\|^2 \|S_\star\| \|X\|^2.$$

We turn our attention to the difference $F(P_\star + X, A_\star, B_\star) - F(P_\star + X, \widehat{A}, \widehat{B})$. We use the notation $P_X$ as a shorthand for $P_\star + X$. Then, by Equation 16 we find

$$F(P_X, \widehat{A}, \widehat{B}, \widehat{Q}) - F(P_X, A_\star, B_\star, Q_\star) = A_\star^\top P_X (I + S_\star P_X)^{-1} A_\star - \widehat{A}^\top P_X \left(I + \widehat{S} P_X\right)^{-1} \widehat{A} - \Delta_Q$$

$$= A_\star^\top P_X (I + S_\star P_X)^{-1} \Delta_S P_X (I + \widehat{S} P_X)^{-1} A_\star - A_\star^\top P_X (I + \widehat{S} P_X)^{-1} \Delta_A$$

$$- \Delta_A^\top P_X (I + \widehat{S} P_X)^{-1} A_\star - \Delta_A^\top P_X (I + \widehat{S} P_X)^{-1} \Delta_A - \Delta_Q. \qquad (28)$$

Then,

$$\|F(P_\star + X, \widehat{A}, \widehat{B}, \widehat{Q}) - F(P_\star + X, A_\star, B_\star, Q_\star)\|$$
$$\leq \|A_\star\|^2 \|P_X\|^2 \|\Delta_S\| + 2\|A_\star\| \|P_X\| \varepsilon + \|P_X\| \varepsilon^2 + \varepsilon,$$

where we used Lemma 7. Since $X \in \mathcal{S}_\nu$, we know $\|X\| \leq \nu$ and hence $\|P_X\| \leq \|P_\star\| + \nu$. We assumed that $\nu \leq 1/2$ and so $\|P_X\| \leq \|P_\star\| + 1$. Now, we know that $\|\Delta_S\| \leq 2\|B_\star\| \|R^{-1}\| \varepsilon + \|R^{-1}\| \varepsilon^2$ and since we assumed $\varepsilon \leq \|B_\star\|$, we have $\|\Delta_S\| \leq 3\|B_\star\| \|R^{-1}\| \varepsilon$. Therefore,

$$\|\Phi(X)\| \leq \frac{\tau(L_\star, \rho)^2}{1 - \rho^2} \left[ \|L_\star\|^2 \|S_\star\| \nu^2 + 3\|A_\star\|_+^2 \|P_\star\|_+^2 \|B_\star\|_+ \|R^{-1}\|_+ \varepsilon \right].$$

We use Lemma 7, the assumption $\nu \leq \|S_\star\|^{-1}$, and the definition of $\mathcal{H}$ to upper bound

$$\|\mathcal{H}(X_1) - \mathcal{H}(X_2)\| \leq \|L_\star\|^2 \left[ \|S_\star\|^2 \nu^2 + 2\|S_\star\| \nu \right] \|X_1 - X_2\| \leq 3\|L_\star\|^2 \|S_\star\| \nu \|X_1 - X_2\|.$$

Let us denote $\mathcal{G}(X) = F(P_\star + X, \widehat{A}, \widehat{B}, \widehat{Q}) - F(P_\star + X, A_\star, B_\star, Q_\star)$. In order to upper bound $\|\mathcal{G}(X_1) - \mathcal{G}(X_2)\|$ we first upper bound the norm of $(I + S_\star P_X)^{-1}$ and $(I + \widehat{S} P_X)^{-1}$. Since $\|X\| \leq \nu \leq 1/2$ and since $P_\star \succeq I$, by Lemma 7 we get

$$\|(I + S_\star P_X)^{-1}\| = \|P_X^{-1} P_X (I + S_\star P_X)^{-1}\| \leq \|P_X^{-1}\| \|P_X (I + S_\star P_X)^{-1}\| \leq 2\|P_X\|.$$

Therefore, after some algebraic manipulations, we obtain

$$\|\mathcal{G}(X_1) - \mathcal{G}(X_2)\| \leq 32\varepsilon \|A_\star\|_+^2 \|P_\star\|_+^3 \|B_\star\|_+^3 \|R^{-1}\|_+^2 \|X_1 - X_2\|.$$

# F   Proofs for LQG

## F.1   Proof of Theorem 3

The proof strategy works as follows. Note that unlike the case for LQR, we were unable to reuse or derive an exact second order perturbation bound analogous to Lemma 3 for LQG. Hence we resort to an argument based on a simple Taylor expansion.

We first assume that $\widehat{K}$ satisfies $\|\widehat{K} - K_\star T^{-1}\| \leq \varepsilon$ and that $\varepsilon \leq 1$. Let $\Theta = (A, B, C, W, V, Q, R)$ be the parameters that specific an instance of LQG in (8). Here, we will be slightly more general than in (8) and allow the process noise $\mathbf{w}_t$ and observation noise $\mathbf{v}_t$ to have non-isotropic covariances $W$ and $V$, respectively. Let $J(\widehat{A}, \widehat{B}, \widehat{C}, \widehat{K}, \widehat{L}; \Theta)$ denote the following cost:

$$J(\widehat{A}, \widehat{B}, \widehat{C}, \widehat{K}, \widehat{L}; \Theta) = \lim_{T \to \infty} \mathbb{E}\left[ \frac{1}{T} \sum_{t=1}^{T} \mathbf{y}_t^\mathsf{T} Q \mathbf{y}_t + \mathbf{u}_t^\mathsf{T} R \mathbf{u}_t \right] \quad \text{s.t.}$$

$$\mathbf{x}_{t+1} = A\mathbf{x}_t + B\mathbf{u}_t + \mathbf{w}_t, \quad \mathbf{w}_t \sim \mathcal{N}(0, W),$$
$$\mathbf{y}_t = C\mathbf{x}_t + \mathbf{v}_t, \quad \mathbf{v}_t \sim \mathcal{N}(0, V),$$
$$\mathbf{u}_t = \widehat{K}\widehat{\mathbf{x}}_t,$$
$$\widehat{\mathbf{x}}_{t+1} = \widehat{A}\widehat{\mathbf{x}}_t + \widehat{B}\mathbf{u}_t + \widehat{L}(\mathbf{y}_t - \widehat{C}\widehat{\mathbf{x}}_t).$$

As observed earlier, we have that $J(\widehat{A}, \widehat{B}, \widehat{C}, \widehat{K}, \widehat{L}; \Theta) = J(\widehat{A}, \widehat{B}, \widehat{C}, \widehat{K}, \widehat{L}, \tilde{\Theta})$, with $\tilde{\Theta} = (TAT^{-1}, TB, CT^{-1}, TWT^\mathsf{T}, V, Q, R)$ for any invertible $T$. Hence, we can assume that $T = I$ moving forward.

We consider the closely related cost $J_s(\widehat{A}, \widehat{B}, \widehat{C}, \widehat{K}, \widehat{L}; \Theta)$ defined as:

$$J_s(\widehat{A}, \widehat{B}, \widehat{C}, \widehat{K}, \widehat{L}; \Theta) = \lim_{T \to \infty} \mathbb{E}\left[ \frac{1}{T} \sum_{t=1}^{T} \mathbf{x}_t^\mathsf{T} Q \mathbf{x}_t + \mathbf{u}_t^\mathsf{T} R \mathbf{u}_t \right] \quad \text{s.t.}$$

$$\mathbf{x}_{t+1} = A\mathbf{x}_t + B\mathbf{u}_t + \mathbf{w}_t, \quad \mathbf{w}_t \sim \mathcal{N}(0, W),$$
$$\mathbf{y}_t = C\mathbf{x}_t + \mathbf{v}_t, \quad \mathbf{v}_t \sim \mathcal{N}(0, V),$$
$$\mathbf{u}_t = \widehat{K}\widehat{\mathbf{x}}_t,$$
$$\widehat{\mathbf{x}}_{t+1} = \widehat{A}\widehat{\mathbf{x}}_t + \widehat{B}\mathbf{u}_t + \widehat{L}(\mathbf{y}_t - \widehat{C}\widehat{\mathbf{x}}_t).$$

Observe that:

$$J(\widehat{A}, \widehat{B}, \widehat{C}, \widehat{K}, \widehat{L}; \Theta) = J_s(\widehat{A}, \widehat{B}, \widehat{C}, \widehat{K}, \widehat{L}; (A, B, C, W, V, Q_c, R)) + \text{tr}(QV), \quad Q_c := C^\mathsf{T} Q C.$$

Let $J_{s,\star}$ denote the optimal value for $J_s$. This decomposition shows that:

$$J(\widehat{A}, \widehat{B}, \widehat{C}, \widehat{K}, \widehat{L}; \Theta) - J_\star = J_s(\widehat{A}, \widehat{B}, \widehat{C}, \widehat{K}, \widehat{L}; (A, B, C, W, V, Q_c, R)) - J_{s,\star}.$$

Hence we will work with $J_s$ from here on. In the sequel, we will drop the dependence of $J$ on $\Theta$. We write $\widehat{\Xi} = (\widehat{A}, \widehat{B}, \widehat{C}, \widehat{K}, \widehat{L})$ and $\Xi_\star = (A, B, C, K_\star, L_\star)$. By Taylor's theorem,

$$J_s(\widehat{\Xi}) - J_{s,\star} = [DJ_s(\Xi_\star)](\Delta) + \frac{1}{2}[D^2 J_s(\tilde{\Xi})](\Delta, \Delta)$$

$$= \frac{1}{2}[D^2 J_s(\tilde{\Xi})](\Delta, \Delta).$$

where the first equality holds by the optimality of $\Xi_\star$ and $\tilde{\Xi}$ is an element along the ray between $\widehat{\Xi}$ and $\Xi_\star$.

We start by defining $\mathbf{e}_t = \widehat{\mathbf{x}}_t - \mathbf{x}_t$. Observe that:

$$\begin{bmatrix} \mathbf{x}_t \\ \widehat{\mathbf{x}}_t \end{bmatrix} = S \begin{bmatrix} \mathbf{x}_t \\ \mathbf{e}_t \end{bmatrix}, \quad S = \begin{bmatrix} I & 0 \\ I & I \end{bmatrix}. \tag{29}$$

We write:

$$\begin{bmatrix} \mathbf{x}_{t+1} \\ \widehat{\mathbf{x}}_{t+1} \end{bmatrix} = \begin{bmatrix} A\mathbf{x}_t + B\widehat{K}\widehat{\mathbf{x}}_t + \mathbf{w}_t \\ \widehat{A}\widehat{\mathbf{x}}_t + \widehat{B}\widehat{K}\widehat{\mathbf{x}}_t + \widehat{L}(C\mathbf{x}_t + \mathbf{v}_t - \widehat{C}\widehat{\mathbf{x}}_t) \end{bmatrix}$$

$$= \begin{bmatrix} A & B\widehat{K} \\ \widehat{L}C & (\widehat{A} + \widehat{B}\widehat{K}) - \widehat{L}\widehat{C} \end{bmatrix} \begin{bmatrix} \mathbf{x}_t \\ \widehat{\mathbf{x}}_t \end{bmatrix} + \begin{bmatrix} I & 0 \\ 0 & \widehat{L} \end{bmatrix} \begin{bmatrix} \mathbf{w}_t \\ \mathbf{v}_t \end{bmatrix}$$

$$=: M(\widehat{A}, \widehat{B}, \widehat{C}, \widehat{K}, \widehat{L}) \begin{bmatrix} \mathbf{x}_t \\ \widehat{\mathbf{x}}_t \end{bmatrix} + \begin{bmatrix} I & 0 \\ 0 & \widehat{L} \end{bmatrix} \begin{bmatrix} \mathbf{w}_t \\ \mathbf{v}_t \end{bmatrix}. \tag{30}$$

We abbreviate $\widehat{M} := M(\widehat{A}, \widehat{B}, \widehat{C}, \widehat{K}, \widehat{L})$. Applying a similarity transform to $\widehat{M}$:

$$S^{-1}\widehat{M}S = \begin{bmatrix} I & 0 \\ -I & I \end{bmatrix} \begin{bmatrix} A & B\widehat{K} \\ \widehat{L}C & (\widehat{A}+\widehat{B}\widehat{K}) - \widehat{L}\widehat{C} \end{bmatrix} \begin{bmatrix} I & 0 \\ I & I \end{bmatrix}$$

$$= \begin{bmatrix} A+B\widehat{K} & B\widehat{K} \\ (\widehat{A}-A) + (\widehat{B}-B)\widehat{K} + \widehat{L}(C-\widehat{C}) & (\widehat{A}-\widehat{L}\widehat{C}) + (\widehat{B}-B)\widehat{K} \end{bmatrix}$$

$$=: \widehat{N} .$$

Therefore $\widehat{M}$ is stable iff $\widehat{N}$ is stable.

By (30), if $\widehat{M}$ is stable, then the stationary distribution of $(\mathbf{x}_t, \widehat{\mathbf{x}}_t)$ is given by $\mathcal{N}(0, \widehat{\Sigma})$ with $\widehat{\Sigma} =$ $\mathsf{dlyap}\left( \widehat{M}^\mathsf{T}, \begin{bmatrix} W & 0 \\ 0 & \widehat{L}V\widehat{L}^\mathsf{T} \end{bmatrix} \right)$. Therefore, we have that:

$$J_s(\widehat{A}, \widehat{B}, \widehat{C}, \widehat{K}, \widehat{L}) = \mathrm{tr}\left( \begin{bmatrix} Q_c & 0 \\ 0 & \widehat{K}^\mathsf{T}R\widehat{K} \end{bmatrix} \widehat{\Sigma} \right) .$$

Now for any $(\tilde{A}, \tilde{B}, \tilde{C}, \tilde{K}, \tilde{L})$, observe we can write $\tilde{M} = M(\tilde{A}, \tilde{B}, \tilde{C}, \tilde{K}, \tilde{L})$ in terms of $M_\star = M(A, B, C, K_\star, L_\star)$ as:

$$\tilde{M} = \tilde{M} - M_\star + M_\star$$

$$= \begin{bmatrix} 0 & B(\tilde{K}-K_\star) \\ (\tilde{L}-L_\star)C & (\tilde{A}-A) + (\tilde{B}-B)\tilde{K} + B(\tilde{K}-K_\star) - ((\tilde{L}-L_\star)\tilde{C} + L_\star(\tilde{C}-C)) \end{bmatrix} + M_\star$$

$$=: \Delta + M_\star .$$

Using the assumption that $\varepsilon \leq 1$, we can bound $\|\Delta\|$ by:

$$\|\Delta\| \leq \|B\|\varepsilon_K + \|C\|\varepsilon_L + \varepsilon_A + (\|K_\star\|+1)\varepsilon_B + \|B\|\varepsilon_K + (\|C\|+1)\varepsilon_L + \|L_\star\|\varepsilon_C$$
$$\leq \varepsilon_A + (\|K_\star\|+1)\varepsilon_B + \|L_\star\|\varepsilon_C + 2(\|C\|+1)\varepsilon_L + 2\|B\|\varepsilon_K$$
$$\leq 2\Gamma_\star(\varepsilon_A + \varepsilon_B + \varepsilon_C + \varepsilon_K + \varepsilon_L)$$
$$\leq 10\Gamma_\star\varepsilon .$$

Let $\rho \geq \rho(M_\star)$. If $10\Gamma_\star \cdot \varepsilon \leq \frac{1-\rho}{2\tau(M_\star,\rho)}$, then we have $\rho(\tilde{M}) \leq (1+\rho)/2$ and $\tau(\tilde{M}, (1+\rho)/2) \leq \tau(M_\star, \rho)$. This holds for any $(\tilde{A}, \tilde{B}, \tilde{C}, \tilde{K}, \tilde{L})$ along the ray of $(\widehat{A}, \widehat{B}, \widehat{C}, \widehat{K}, \widehat{L})$ and $(A, B, C, K, L)$. Therefore for any point along this ray we have that $\tilde{M}$ is stable and that for any fixed $H$, $\|\mathsf{dlyap}(\tilde{M}^\mathsf{T}, H)\| \leq \frac{\|H\|\tau(M_\star,\rho)}{1-\mathsf{Avg}(\rho,1)^2}$.

Next, we consider $\tilde{\Xi} = (\tilde{A}, \tilde{B}, \tilde{C}, \tilde{K}, \tilde{L})$ and differentiate $J_s$ twice w.r.t. $\tilde{\Xi}$, assuming that $\tilde{\Xi}$ corresponds to a point along the ray $\widehat{\Xi} = (\widehat{A}, \widehat{B}, \widehat{C}, \widehat{K}, \widehat{L})$ and $\Xi_\star = (A, B, C, K_\star, L_\star)$. By the chain rule:

$$[D^2 J_s(\tilde{\Xi})](\Delta, \Delta)$$
$$= \mathrm{tr}\left( \begin{bmatrix} Q_c & 0 \\ 0 & \tilde{K}^\mathsf{T}R\tilde{K} \end{bmatrix} [D^2\Sigma(\tilde{\Xi})](\Delta, \Delta) \right) + \mathrm{tr}\left( \begin{bmatrix} 0 & 0 \\ 0 & \Delta_K^\mathsf{T}R\tilde{K} + \tilde{K}^\mathsf{T}R\Delta_K \end{bmatrix} [D\Sigma(\tilde{\Xi})](\Delta) \right)$$
$$\quad + 2\,\mathrm{tr}\left( \begin{bmatrix} 0 & 0 \\ 0 & \Delta_K^\mathsf{T}R\Delta_K \end{bmatrix} \Sigma(\tilde{\Xi}) \right)$$
$$\leq (\mathrm{tr}(Q_c) + \mathrm{tr}(R)(\|K_\star\|+1)^2)\|[D^2\Sigma(\tilde{\Xi})](\Delta, \Delta)\|$$
$$\quad + 2\,\mathrm{tr}(R)(\|K_\star\|+1)\varepsilon_K\|[D\Sigma(\tilde{\Xi})](\Delta)\| + 2\,\mathrm{tr}(R)\varepsilon_K^2\|\Sigma(\tilde{\Xi})\|$$
$$\leq (\mathrm{tr}(Q_c) + \mathrm{tr}(R)\Gamma_\star^2)\left( \|[D^2\Sigma(\tilde{\Xi})](\Delta, \Delta)\| + 2\varepsilon_K\|[D\Sigma(\tilde{\Xi})](\Delta)\| + 2\varepsilon_K^2\|\Sigma(\tilde{\Xi})\| \right) .$$

Given the composition $h(x) = g(f(x))$, we have by the chain rule:

$$[Dh(x)](\Delta) = [Dg(f(x))]([Df(x)](\Delta)) , \tag{31}$$
$$[D^2 h(x)](\Delta, \Delta) = [D^2 g(f(x))]([Df(x)](\Delta), [Df(x)](\Delta)) + [Dg(f(x))]([D^2 f(x)](\Delta, \Delta)) . \tag{32}$$

We apply this with $g(M, Q) = \mathsf{dlyap}(M, Q)$ and

$$f(\tilde{\Xi}) = \left( \begin{bmatrix} A & B\tilde{K} \\ \tilde{L}C & (\tilde{A} + \tilde{B}\tilde{K}) - \tilde{L}\tilde{C} \end{bmatrix}^{\mathsf{T}}, \begin{bmatrix} W & 0 \\ 0 & \tilde{L}V\tilde{L}^{\mathsf{T}} \end{bmatrix} \right) .$$

Therefore the composition $h = g \circ f$ is $h(\Xi) = \Sigma(\Xi)$. Differentiating $f$,

$$[Df(\tilde{\Xi})](\Delta) = \left( \begin{bmatrix} 0 & B\Delta_K \\ \Delta_L C & \Delta_A + \tilde{B}\Delta_K + \Delta_B\tilde{K} - (\tilde{L}\Delta_C + \Delta_L\tilde{C}) \end{bmatrix}^{\mathsf{T}}, \begin{bmatrix} 0 & 0 \\ 0 & \tilde{L}V\Delta_L^{\mathsf{T}} + \Delta_L V\tilde{L}^{\mathsf{T}} \end{bmatrix} \right) ,$$

$$[D^2 f(\tilde{\Xi})](\Delta, \Delta) = \left( \begin{bmatrix} 0 & 0 \\ 0 & 2\Delta_B\Delta_K - 2\Delta_L\Delta_C \end{bmatrix}^{\mathsf{T}}, \begin{bmatrix} 0 & 0 \\ 0 & 2\Delta_L V\Delta_L^{\mathsf{T}} \end{bmatrix} \right) .$$

Letting $P = g(M, Q)$, we have:

$$[Dg(M, Q)](\Delta) = g(M, \Delta_Q + M^{\mathsf{T}}P\Delta_M + \Delta_M^{\mathsf{T}}PM) ,$$

$$[D^2 g(M, Q)](\Delta, \Delta) = 2g(M, \Delta_M^{\mathsf{T}}P\Delta_M + M^{\mathsf{T}}g(M, \Delta_Q)\Delta_M + \Delta_M^{\mathsf{T}}g(M, \Delta_Q)M)$$
$$+ 2g(M, \Delta_M^{\mathsf{T}}g(M, \Delta_M^{\mathsf{T}}PM + M^{\mathsf{T}}P\Delta_M)M + M^{\mathsf{T}}g(M, \Delta_M^{\mathsf{T}}PM + M^{\mathsf{T}}P\Delta_M)\Delta_M) .$$
(33)

The derivation for (33) is postponed for now.

Let $\mathcal{S}_M(Q)$ be the linear operator $Q \mapsto g(M, Q)$. Observe that $\|\mathcal{S}_M\| \leq \frac{\tau^2(M,\rho)}{1-\rho^2}$ and $\|M^{\mathsf{T}}g(M, Q)\| \leq \frac{\tau^2(M,\rho)}{1-\rho^2}\|Q\|$. With this, we can bound,

$$\|[Dg(M, Q)](\Delta)\| \leq \frac{\tau^2(M, \rho)}{1 - \rho^2}\|\Delta_Q\| + 2\frac{\tau^4(M, \rho)}{(1 - \rho^2)^2}\|Q\|\|\Delta_M\| ,$$

$$\|[D^2 g(M, Q)](\Delta, \Delta)\| \leq 10\frac{\tau^6(M, \rho)}{(1 - \rho^2)^3}\|Q\|\|\Delta_M\|^2 + 4\frac{\tau^4(M, \rho)}{(1 - \rho^2)^2}\|\Delta_Q\|\|\Delta_M\| .$$

Therefore,

$$\|[Dg(f(\tilde{\Xi}))]([Df(\tilde{\Xi})](\Delta))\|$$

$$\leq 2\frac{\tau^2(\tilde{M}, \tilde{\rho})}{1 - \tilde{\rho}^2}\|\tilde{L}\|\|V\|\varepsilon_L + 20\frac{\tau^4(\tilde{M}, \tilde{\rho})}{(1 - \tilde{\rho}^2)^2}\max\{\|W\|, \|\tilde{L}V\tilde{L}^{\mathsf{T}}\|\}\Gamma_\star\varepsilon$$

$$\leq 2\frac{\tau^2(M_\star, \rho)}{1 - \mathsf{Avg}(\rho, 1)^2}\Gamma_\star\|V\|\varepsilon_L + 20\frac{\tau^4(M_\star, \rho)}{(1 - \mathsf{Avg}(\rho, 1)^2)^2}\max\{\|W\|, \Gamma_\star^2\|V\|\}\Gamma_\star\varepsilon$$

$$\leq 22\frac{\tau^4(M_\star, \rho)}{(1 - \mathsf{Avg}(\rho, 1)^2)^2}\max\{\|W\|, \|V\|\}\Gamma_\star^3\varepsilon .$$

and,

$$\|[Dg(f(\tilde{\Xi}))]([D^2 f(\tilde{\Xi})](\Delta, \Delta))\|$$

$$\leq 2\frac{\tau^2(\tilde{M}, \tilde{\rho})}{1 - \tilde{\rho}^2}\|V\|\varepsilon_L^2 + 4\frac{\tau^4(\tilde{M}, \tilde{\rho})}{(1 - \tilde{\rho}^2)^2}\max\{\|W\|, \|\tilde{L}V\tilde{L}^{\mathsf{T}}\|\}(\varepsilon_B\varepsilon_K + \varepsilon_L\varepsilon_C)$$

$$\leq 2\frac{\tau^2(M_\star, \rho)}{1 - \mathsf{Avg}(\rho, 1)^2}\|V\|\varepsilon_L^2 + 8\frac{\tau^4(M_\star, \rho)}{(1 - \mathsf{Avg}(\rho, 1)^2)^2}\max\{\|W\|, \Gamma_\star^2\|V\|\}\varepsilon^2$$

$$\leq 10\frac{\tau^4(M_\star, \rho)}{(1 - \mathsf{Avg}(\rho, 1)^2)^2}\max\{\|W\|, \|V\|\}\Gamma_\star^2\varepsilon^2 .$$

and also,

$$\|[D^2 g(f(\tilde{\Xi}))]([Df(\tilde{\Xi})](\Delta), [Df(\tilde{\Xi})](\Delta))\|$$

$$\leq 10\frac{\tau^6(\tilde{M}, \tilde{\rho})}{(1 - \tilde{\rho}^2)^3}\max\{\|W\|, \|\tilde{L}V\tilde{L}^{\mathsf{T}}\|\}(10\Gamma_\star\varepsilon)^2 + 4\frac{\tau^4(\tilde{M}, \tilde{\rho})}{(1 - \tilde{\rho}^2)^2}(2\|\tilde{L}\|\|V\|\varepsilon_L)(10\Gamma_\star\varepsilon)$$

$$\leq 1000\frac{\tau^6(M_\star, \rho)}{(1 - \mathsf{Avg}(\rho, 1)^2)^3}\max\{\|W\|, \|V\|\}\Gamma_\star^4\varepsilon^2 + 80\frac{\tau^4(M_\star, \rho)}{(1 - \mathsf{Avg}(\rho, 1)^2)^2}\|V\|\Gamma_\star\varepsilon^2$$

$$\leq 1080\frac{\tau^6(M_\star, \rho)}{(1 - \mathsf{Avg}(\rho, 1)^2)^3}\max\{\|W\|, \|V\|\}\Gamma_\star^4\varepsilon^2 .$$

Therefore,

$$\|\Sigma(\tilde{\Xi})\| \leq \frac{\tau^2(M_\star, \rho)}{1 - \mathsf{Avg}(\rho, 1)^2} \max\{\|W\|, \|V\|\}\Gamma_\star^2,$$

$$\|[D\Sigma(\tilde{\Xi})](\Delta)\| \leq 22\frac{\tau^4(M_\star, \rho)}{(1 - \mathsf{Avg}(\rho, 1)^2)^2} \max\{\|W\|, \|V\|\}\Gamma_\star^3\varepsilon,$$

$$\|[D^2\Sigma(\tilde{\Xi})](\Delta, \Delta)\| \leq 1090\frac{\tau^6(M_\star, \rho)}{(1 - \mathsf{Avg}(\rho, 1)^2)^3} \max\{\|W\|, \|V\|\}\Gamma_\star^4\varepsilon^2.$$

Combining these bounds with the calculations for $J_s$,

$$[D^2 J_s(\tilde{\Xi})](\Delta, \Delta)$$
$$\leq (\mathrm{tr}(Q_c) + \mathrm{tr}(R)\Gamma_\star^2) \left(\|[D^2\Sigma(\tilde{\Xi})](\Delta, \Delta)\| + 2\varepsilon_K\|[D\Sigma(\tilde{\Xi})](\Delta)\| + 2\varepsilon_K^2\|\Sigma(\tilde{\Xi})\|\right)$$
$$\leq 1136(\mathrm{tr}(Q_c) + \mathrm{tr}(R)\Gamma_\star^2)\frac{\tau^6(M_\star, \rho)}{(1 - \mathsf{Avg}(\rho, 1)^2)^3} \max\{\|W\|, \|V\|\}\Gamma_\star^4\varepsilon^2$$
$$\leq 1136(\mathrm{tr}(Q_c) + \mathrm{tr}(R))\frac{\tau^6(M_\star, \rho)}{(1 - \mathsf{Avg}(\rho, 1)^2)^3} \max\{\|W\|, \|V\|\}\Gamma_\star^6\varepsilon^2$$
$$\leq 1136 \cdot 64(\mathrm{tr}(Q_c) + \mathrm{tr}(R))\frac{\tau^6(M_\star, \rho)}{(1 - \rho^2)^3} \max\{\|W\|, \|V\|\}\Gamma_\star^6\varepsilon^2.$$

To finish the first part of proof, we show the derivation for (33). Observe that for $E$ small:

$$(A + E)^{-1} = (A(I + A^{-1}E))^{-1} = (I + A^{-1}E)^{-1}A^{-1} = \sum_{k=0}^{\infty}(-A^{-1}E)^k A^{-1}$$
$$= A^{-1} - A^{-1}EA^{-1} + A^{-1}EA^{-1}EA^{-1} + O(\|E\|^3).$$

Therefore, letting $S = I - M^\mathsf{T} \otimes M^\mathsf{T}$,

$$(I - (M + \Delta_M)^\mathsf{T} \otimes (M + \Delta_M)^\mathsf{T})^{-1}\mathrm{vec}(Q + \Delta_Q)$$
$$= S^{-1}\mathrm{vec}(Q + \Delta_Q)$$
$$\quad + S^{-1}(\Delta_M^\mathsf{T} \otimes M^\mathsf{T} + M^\mathsf{T} \otimes \Delta_M^\mathsf{T} + \Delta_M^\mathsf{T} \otimes \Delta_M^\mathsf{T})S^{-1}\mathrm{vec}(Q + \Delta_Q)$$
$$\quad + S^{-1}(\Delta_M^\mathsf{T} \otimes M^\mathsf{T} + M^\mathsf{T} \otimes \Delta_M^\mathsf{T})S^{-1}(\Delta_M^\mathsf{T} \otimes M^\mathsf{T} + M^\mathsf{T} \otimes \Delta_M^\mathsf{T})S^{-1}\mathrm{vec}(Q + \Delta_Q) + O(\|\Delta\|^3)$$
$$= S^{-1}\mathrm{vec}(Q) + S^{-1}\mathrm{vec}(\Delta_Q)$$
$$\quad + S^{-1}(\Delta_M^\mathsf{T} \otimes M^\mathsf{T} + M^\mathsf{T} \otimes \Delta_M^\mathsf{T})S^{-1}\mathrm{vec}(Q) + S^{-1}(\Delta_M^\mathsf{T} \otimes \Delta_M^\mathsf{T})S^{-1}\mathrm{vec}(Q)$$
$$\quad + S^{-1}(\Delta_M^\mathsf{T} \otimes M^\mathsf{T} + M^\mathsf{T} \otimes \Delta_M^\mathsf{T})S^{-1}\mathrm{vec}(\Delta_Q)$$
$$\quad + S^{-1}(\Delta_M^\mathsf{T} \otimes M^\mathsf{T} + M^\mathsf{T} \otimes \Delta_M^\mathsf{T})S^{-1}(\Delta_M^\mathsf{T} \otimes M^\mathsf{T} + M^\mathsf{T} \otimes \Delta_M^\mathsf{T})S^{-1}\mathrm{vec}(Q) + O(\|\Delta\|^3).$$

Now to finish the proof, we need to obtain a $\overline{\varepsilon}$ such that

$$\max\{\|\widehat{A} - TA_\star T^{-1}\|, \|\widehat{B} - TB_\star\|, \|\widehat{C} - C_\star T^{-1}\|, \|\widehat{L} - TL_\star\|, \|\widehat{K} - K_\star T^{-1}\|\} \leq \overline{\varepsilon}.$$

We do this by observing that:

$$K_\star T^{-1} = \mathsf{LQR}(TA_\star T^{-1}, TB_\star, T^{-\mathsf{T}}C_\star^\mathsf{T}QC_\star T^{-1}, R).$$

Hence with $\widehat{K} = \mathsf{LQR}(\widehat{A}, \widehat{B}, \widehat{C}^\mathsf{T}Q\widehat{C}, R)$, we can use the Riccati perturbation between $\overline{P}_\star = \mathsf{dare}(TA_\star T^{-1}, TB_\star, T^{-\mathsf{T}}C_\star^\mathsf{T}QC_\star T^{-1}, R)$ and $\widehat{P} = \mathsf{dare}(\widehat{A}, \widehat{B}, \widehat{C}^\mathsf{T}Q\widehat{C}, R)$. We can bound, assuming that that $\varepsilon_C \leq \|C_\star\|$:

$$\|T^{-\mathsf{T}}C_\star^\mathsf{T}QC_\star T^{-1} - \widehat{C}^\mathsf{T}Q\widehat{C}\| \leq (2\|C_\star\| + \varepsilon_C)\|Q\|\varepsilon_C \leq 3\|C_\star\|\|Q\|\varepsilon_C.$$

Therefore by our hypothesis and the assumption that $f$ is monotonically increasing,

$$\|\widehat{P} - \overline{P}_\star\| \leq f(\max\{\varepsilon_A, \varepsilon_B, 3\|C_\star\|\|Q\|\varepsilon_C\}) \leq f(3\|C_\star\|_+\|Q\|_+ \max\{\varepsilon_A, \varepsilon_B, \varepsilon_C\}).$$

as long as $3\|C_\star\|_+\|Q\|_+ \max\{\varepsilon_A, \varepsilon_B, \varepsilon_C\} \le \gamma_0$. By Lemma 2, this allows us to bound:

$$\|\widehat{K} - K_\star T^{-1}\| \le \frac{7\Gamma_\star^3}{\underline{\sigma}(R)} \max\{\varepsilon_A, \varepsilon_B, f(3\|C_\star\|_+\|Q\|_+ \max\{\varepsilon_A, \varepsilon_B, \varepsilon_C\})\}$$

$$\le \frac{7\Gamma_\star^3}{\underline{\sigma}(R)} f(3\|C_\star\|_+\|Q\|_+ \max\{\varepsilon_A, \varepsilon_B, \varepsilon_C\}) .$$

We can set $\overline{\varepsilon} = \frac{7\Gamma_\star^3}{\underline{\sigma}(R)} f(3\|C_\star\|_+\|Q\|_+ \max\{\varepsilon_A, \varepsilon_B, \varepsilon_C\})$ because we assume that $f(\gamma) \ge \gamma$ and $\underline{\sigma}(R) \ge 1$.

We now complete the proof by observing that for $M_\star = \begin{bmatrix} A_\star & B_\star K_\star \\ L_\star C_\star & A_\star + B_\star K_\star - L_\star C_\star \end{bmatrix}$ and $N_\star = \begin{bmatrix} A_\star + B_\star K_\star & B_\star K_\star \\ 0 & A_\star - L_\star C_\star \end{bmatrix}$, using $S$ defined in (29), we have that $N_\star = S^{-1} M_\star S$. This identity has several consequences. First, we have $\rho(N_\star) = \rho(M_\star)$. Second, for any integer $k \ge 1$ we have $S^{-1} M_\star^k S = N_\star^k$, and therefore

$$\|N_\star^k\| \le \mathrm{cond}(S)\|M_\star^k\| = \frac{3 + \sqrt{5}}{2}\|M_\star^k\| .$$

A nearly identical argument shows that $\|M_\star^k\| \le \frac{3+\sqrt{5}}{2}\|N_\star^k\|$. Therefore, $\tau(M_\star, \rho) \asymp \tau(N_\star, \rho)$ for any $\rho$.