[Reviews · NeurIPS 2019]

Reviewer 1



1. Originality: The paper considers standard control setting in LQR and LQG with unknown dynamics, and standard algorithm---certainty equivalence. A key part of the analysis is based on perturbation bounds for discrete Riccati equations, which uses an existing results from Konstantinov et al (the authors provided a new analysis but at the cost of a stronger assumption). Finite sample analysis of certainty equivalence method is not new in RL literature, especially for tabular MDPs. In my perspective, the survey on related works on sample complexity of certainty equivalence methods for different models other than LQR should be included in related work for discussion. 2. Quality: the submission is technically sound and the results are well supported by theoretical statements. The authors also carefully compared their results to previous results using a robust control approach to account model uncertainty. The result is that certainty equivalence method could perform better under the regime of small model estimation error, but could be more sensitive to model error. Would it be possible to conduct experiments to verify and to better illustrate the tradeoff between these two? 3. clarity: I think the paper is well written and organized. 4. Significance: The finite sample analysis of certainty equivalence in LQR and LQG was missing previously and the paper fills this gaps. While the analysis techniques used in paper existed before (e.g., Riccati Perturbation Theory), the final result is new to me.

Reviewer 2



The result of theorems 1, 2 and 3 hold when the error in the parameter estimation is “sufficiently” small. However, the regime that these theorems allow for estimation error is almost never practical. For example, Theorem 2 holds as long as the right hand side is smaller than \sigma^2_w. Consider a system with d=l=4 and \beta = \Gamma^* = \tau=2. This requires \varepsilon=2^{-23} which is not possible for most practical purposes. As the comparison in Section 2.1 suggests, the constants behind the main results of this paper are significantly larger than that of Theorem 4.1 of Dean et. al (2017). Despite the faster rate on , the price paid for the constants is too much to call it an improvement upon Dean et. al (2017). In the online control setting, it seems that the idea of using the offline algorithm with \epsilon-greedy exploration should work, it would be nice if the authors can provide a more rigorous statement and proof for Corollary 1. It would also have been useful to see some numerical results to get a better sense of the theoretical results. Minor Comments: I couldn’t find the proof of Theorem 2. Also, I suggest to write the exact universal constant for Theorem 2 and 3 rather than using O(1) notation. Page 2, line 72: \underline{\sigma} is not defined upto this point. I think the first time it is defined is line 105. In Assumption 2, it is assumed that the system is (l,\vartheta)-controllable. but what are (l,\vartheta)? Page 6, line 232: What is “dare”? I don’t think this is defined before.

Reviewer 3



This paper studies the performance of certainty equivalence principle, i.e., fitting a model from given data and designing a controller using the fitted model, in Linear Quadratic systems. Originality: The results rely critically on the extension of an existing result in perturbation theory of discrete Riccati equations. The bounds reflect explicit dependence on system level parameters, unlike previous literature. Quality and Clarity: I have checked some of the proofs and they seem correct to me. The paper is easy to follow, and the statement of the main results make it clear to follow the next steps. Significance: I feel that the contributions are reasonably significant to be published at NeurIPS. The main reason is that it connects seamlessly to past work on system identification.

[Author Response · NeurIPS 2019]

We thank the reviewers for their detailed feedback, which will improve the presentation of our paper.

First we would like to address a high level point that was raised by the reviewers regarding Riccati perturbations. While the proof of Proposition 1 follows the argument of Konstantinov et al., Proposition 2 relies on a new elementary proof technique that is of independent interest and offers a tighter guarantee for certain classes of problems, as discussed in Lines 268-275.

**Reviewer 4.**

- Regarding the rich literature on certainty equivalence for tabular MDPs, we did not discuss tabular MDPs because our focus was on LQR. Nevertheless, we agree with the reviewer that the tabular MDP literature is relevant and should be included; we will introduce a few paragraphs on related works (such as Azar et al. 2013) in our revision. We would appreciate suggestions of the most relevant works studying certainty equivalence for tabular MDPs.

- It is definitely possible to verify empirically that certainty equivalence outperforms robust control methods when the model estimation error is small, while being more sensitive to the size of the error. Dean et al. [2017] observed exactly this in Figure 2; we offer a theoretical justification for their observations.

**Reviewer 5.**

- At this stage, our results do not offer bounds that can be used numerically in practice. Indeed, as the reviewer suggests, they would be too conservative. However, our results offer insights about the performance of certainty equivalence for LQR and LQG. For example, it explains why Dean et al. [2017] observed empirically (c.f. Figure 2 of their paper) that certainty equivalence performs poorly in the high error regime, but outperforms their robust methods in the low error regime.

- Dean et al. [2017] do not analyze certainty equivalence, a popular method used in practice. Moreover, Dean et al. [2017] study only fully observed systems, i.e. they studied LQR, but not LQG. We showed that certainty equivalence achieves a fast statistical rate for both LQR and LQG. We would like to emphasize that the partially observed case is significantly more challenging than the fully observed case.

- Theorem 2 follows from plugging in the inequality from Proposition 2 into the bound of Theorem 1. We will make this more clear in our revision.

- We will make sure to define our notation before it is used in the revision; $\mathrm{dare}(A, B, Q, R)$ is the unique positive semidefinite solution to the discrete algebraic Riccati equation associated with the parameters $A$, $B$, $Q$, and $R$.

- The parameters $(\ell, \nu)$ quantify how controllable a system is. They are system dependent quantities. For example, if the system is controllable in the classical sense, one can choose $\ell$ to be equal to the state dimension and $\nu$ to be the minimal singular value of the controllability matrix. This is not the only choice, however, and this additional degree of freedom allows us to offer sharper bounds for certain systems (c.f. Lines 268-275).

**Reviewer 6.**

- Lower bounds would indeed be very desirable, but unfortunately so far we have not been able to derive lower bounds. We leave this for future work.

- Cohen et al. [2019] focus on the online and fully observed setting and offer an elegant method based on semidefinite programming which achieves $\sqrt{T}$ regret, where $T$ is the horizon. However, their method requires the initialization of the system parameters to be in an error ball of radius $\mathcal{O}(1/\sqrt[4]{T})$, while our method does not have this restriction. Our method also offers computational advantages over Cohen et al. [2019], since we can take advantage of specialized DARE solvers instead of relying on general SDP solvers. We will make this comparison to the work of Cohen et al. [2019] explicit.

- It would indeed be valuable to derive bounds which rely only on observed data and not on unknown problem dependent quantities. At this time we are not aware of a way to achieve this.

# References

A. Cohen, T. Koren, and Y. Mansour. Learning Linear-Quadratic Regulators Efficiently with only $\sqrt{T}$ Regret. *arXiv:1902.06223*, 2019.

S. Dean, H. Mania, N. Matni, B. Recht, and S. Tu. On the Sample Complexity of the Linear Quadratic Regulator. *arXiv:1710.01688*, 2017.


[Meta-Review · NeurIPS 2019]

The paper studies the certainty-equivalence (aka plug-in) principle for linear quadratic systems (LQR and LQG). The main result is a "fast rate" style of guarantee, estimating model parameters to accuracy epsilon gives a policy error of O(epsilon^2). There are several things to like here: results for LQG, a deeper understanding of plug-in estimators, and a honest/detailed comparison to prior work. One concern brought up by the reviewers is that the pre-multipliers in the results are relatively large (in particular larger than those in Dean et al.) and difficult to interpret. The paper is honest about this in the discussion and a larger constant factor is confirmed by prior empirical results, so I do not view this as a deal-breaker. (As an aside, it might be nice to include that experiment in this paper, just so it is more immediately accessible to readers.)